# Mechanics of blunting of actin–myosin interaction dynamics by the actinopathy-causing mutation E334Q in cytoskeletal $\gamma$-actin

Irene Pertici[1] (ID), Valentina Buonfiglio[1] (ID), Johannes N. Greve[2] (ID), Elena Battirossi[1] (ID), Duccio Fanelli[3], Dietmar J. Manstein[2,4] (ID) and Pasquale Bianco[1] (ID)

[1] *PhysioLab, Department of Biology, University of Florence, Sesto Fiorentino, Italy*
[2] *Institute for Biophysical Chemistry, Hannover Medical School, Fritz-Hartmann-Centre for Medical Research, Hannover, Germany*
[3] *Department of Physics and Astronomy, University of Florence, Sesto Fiorentino, Italy*
[4] *Division for Structural Biochemistry, Hannover Medical School, Hannover, Germany*

Handling Editors: Paul Greenhaff & Wolfgang Linke

The peer review history is available in the Supporting Information section of this article (https://doi.org/10.1113/JP289622#support-information-section).

<div style="writing-mode: vertical-rl">The Journal of Physiology</div>

**Abstract figure legend** Residue E334 is located in a region of the cytoskeletal $\gamma$-actin monomer that is fundamental for the interaction with myosin. In humans, its mutation p.E334Q has been found to lead to several phenotypic alterations

mostly affecting cerebral cortex development and muscle tone, and generically classified under the group of rare diseases named non-muscle actinopathies (NMAs). However, the molecular and cellular alterations caused by the mutation underlying the pathological phenotype are still unclear and complicated by cell remodelling. The effect of the mutation on the actin–myosin interaction dynamics has been dissected by means of a synthetic nanomachine, in which an array of myosin motors interacts with a single actin filament in 2 mM ATP under the loading conditions experienced *in situ*. By employing three different filamentous actin models [wild-type (WT), mutant (p.E334Q) and a hybrid heterofilament (50% WT and 50% p.E334Q monomers)], we demonstrate a drastic impairment of the ability to generate force and shortening by the $\gamma$-actin–myosin interaction. Definition of the blunting of the basic function caused by the E334Q $\gamma$-actin mutation serves as a starting point for understanding the chain of remodelling events leading to the pathology and developing targeted pharmacological interventions.

**Abstract** Cellular processes such as cytokinesis, apoptosis and migration rely heavily on the myosin-based contractility of the $\gamma$-actin network in the submembrane cortex. Direct measurements of $\gamma$-actin–myosin interactions through morphological and depletion investigations remain elusive. Here, we use a synthetic nanomachine, consisting of an array of myosin motors carried on a nanopositioner and brought to interact with an actin filament attached to a bead trapped in the focus of dual laser optical tweezers. The nanomachine is able to mimic the loading conditions of $\gamma$-actin–myosin interactions *in situ*, allowing measurements of the maximum steady force ($F_0$) and of the shortening velocity against loads $< F_0$. Comparative measurements are conducted on wild-type $\gamma$-actin and $\gamma$-actin carrying the E334Q mutation, associated with non-muscle actinopathies. Our results show that the force of the single actin–myosin interaction is 2.5 pN for the wild-type actin and is halved by the mutation. The kinetics of motor attachment–detachment, underpinning the rate of isometric force rise and the force–velocity relation, are also reduced by a factor of two, resulting in a reduction of the maximum nanomachine power to one-fifth. The identification and quantitative definition of the loss of basic function caused by the E334Q $\gamma$-actin mutation serve as a starting point for understanding the chain of remodelling events leading to the pathological phenotype and demonstrate the potential of the nanomachine for targeted therapeutic interventions.

(Received 7 July 2025; accepted after revision 27 October 2025; first published online 12 November 2025)

**Corresponding author** I. Pertici: PhysioLab, Department of Biology, University of Florence, 50019 Sesto Fiorentino, Italy. Email: irene.pertici@unifi.it

### Key points

- Mutations in cytoskeletal actin cause rare pathologies classified as non-muscle actinopathies (NMAs), including the Baraitser–Winter cerebrofrontofacial syndrome characterized by neural cortex abnormalities leading to facial dysmorphism, developmental delay and organ malformations.
- Cytoskeleton dynamics control cell morphology and migration and rely on the interaction of non-muscle myosin II with cytoskeletal $\gamma$-actin, but the system's mechanical performance and its blunting by NMA-causing mutations in $\gamma$-actin have still to be defined.
- Here, a synthetic nanomachine is used to record the relevant mechano-kinetic parameters of the $\gamma$-actin–myosin interaction under physiological loading conditions.
- Quantitative estimates of these parameters for wild-type and E334Q mutant $\gamma$-actin suggest that strong defects in the actin–myosin interaction mechanics may be one of the main molecular mechanisms leading to the pathological phenotype.
- This paper lays the groundwork for the quantitative definition of the basic function altered by NMA-causing mutations and the evaluation of targeted therapies.

## Introduction

The maintenance of cell shape and plasticity as well as cytokinesis, apoptosis and cell locomotion rely on the interaction between non-muscle myosin II filaments and cytoskeletal actin filaments constituted by the $\beta$- and $\gamma$-actin isoforms. Actin isoforms are codified by six genes sharing high sequence similarity ($> 90\%$). While four of these (*ACTA1*, *ACTA2*, *ACTC1* and *ACTG2*) code for muscle tissue-specific actin isoforms, *ACTB* and *ACTG1* are ubiquitously expressed in the human body and code for the cytoskeletal $\beta$- and $\gamma$-actin isoforms, respectively (Vandekerckhove & Weber, 1978). Notably, cytoskeletal $\beta$- and $\gamma$-actin sequences differ by only four out of 375 amino acids, all of which are located at the N-terminus.

The spatial subcellular localization of the $\beta$- and $\gamma$-isoforms has been a subject of controversy, with a complex and not fully understood picture emerging from various studies (Dugina et al., 2009; Otey et al., 1986; Shestakova et al., 2001). However, a consensus has been reached that these cytoskeletal actins have distinct and non-redundant biological roles despite their high structural similarity. While involvement of $\beta$-actin in cell motility seems to be well established due to its preferential localization in myosin-containing stress fibres, which promote cell adhesion and tail retraction (Tojkander et al., 2012), the role of $\gamma$-actin is more elusive, which is the reason for our interest in the application of the myosin-based nanomachine (Pertici et al., 2018) to define the performance of $\gamma$-actin in loaded motility events. By using newly generated monoclonal antibodies in stationary and migrating cells, Dugina et al. (2009) reported an overall dynamic organization for this isoform characterized by a distinct spatio-temporal regulation depending on cellular activity. In fibroblasts, $\gamma$-actin is predominantly found at the cortex, organized in the form of a microfilament meshwork underlying the plasma membrane, and in non-contractile dorsal stress fibres, which allow force transmission to the competent substrate. In polarized epithelial cells, $\gamma$-actin is mainly localized under the apical membrane. During cell motility, protrusive lamellipodia appear enriched in $\gamma$-actin, supporting a role for this isoform in the control of directional motility.

The preferential location at the cortex and the ability to generate well-entangled contractile networks suggest a prominent role of $\gamma$-actin in maintaining cell morphology, facilitating cell shape flexibility and promoting cell migration. 3D cell migration involves a contractile apparatus at the cell cortex characterized by the formation of blister-like membrane protrusions (blebs) and retractions (Charras & Paluch, 2008; Charras et al., 2006). Bleb swelling is initiated by an increase in intracellular pressure, leading to plasma membrane separation from the cortical cytoskeleton. This is followed by actin polymerization, which ultimately leads to interaction between myosin and actin, powering bleb retraction (Brito & Sousa, 2020).

Cytoskeleton dynamics occurs by interaction of cytoskeletal actin with non-muscle myosin II (NM II), which, in non-muscle cells, self-associates to form ∼300 nm long bipolar filaments (Billington et al., 2013). $\gamma$-Actin has been shown to interact with the two major non-muscle myosin isoforms, NM IIA and NM IIB, stimulating myosin ATPase activity to a similar extent as $\beta$-actin, and four times greater than that measured in the presence of skeletal muscle $\alpha$-actin (Müller et al., 2013). Notably, $\gamma$-actin networks exhibit increased stiffness and larger contraction foci after myosin addition, suggesting that $\gamma$-actin can form stronger interactions with myosin motors and promote greater contractility than $\beta$-actin (Nietmann et al., 2023).

Mutations in cytoskeletal actins lead to a broad range of rare pathological phenotypes, collectively classified as non-muscle actinopathies (NMAs). The majority of the missense mutations in *ACTB* and *ACTG1* are associated with Baraitser–Winter cerebrofrontofacial (BWCFF) syndrome, characterized by cortical abnormalities leading to recognizable facial dysmorphism, developmental delay and organ malformations (Di Donato et al., 2014, 2016). Over 50 missense mutations are reported for *ACTG1* (Parker et al., 2020), with nearly half of these associated with BWCFF, although they often exhibit a milder phenotype (Di Donato et al., 2014, 2016; Parker et al., 2020). One notable example is the point mutation E334Q in $\gamma$-actin, which has been linked to a rare form of NMA. This mutation is particularly interesting because it does not fit into the typical BWCFF-associated

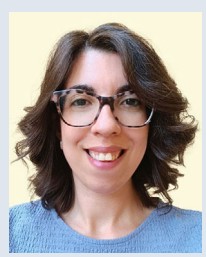

**Irene Pertici** is a fixed-term researcher (RTD-A) in Physiology at the Department of Biology of the University of Florence, Italy. In 2018 she received her PhD in Molecular Medicine by the University of Siena, Italy. Within the research group PhysioLab at the University of Florence she contributed to the development of a synthetic myosin-based nanomachine able to mimic the generation of steady force and shortening of striated muscle. Her main interests focus on defining the mechanisms underlying the function and dysfunction of actin–myosin performance, integrating *in vitro* and *in situ* approaches.

phenotype, and its features are not well characterized. A patient carrying the E334Q mutation was reported to have mild facial anomalies and dysgyria, and to have speech delay and severe muscle hypotonia (Di Donato et al., 2016). Establishing a causal association between a single mutation in cytoskeletal actin and a particular pathology is a complex issue, as it is often complicated by the dynamic nature of actin remodelling. To address this challenge, a crucial step is to reproduce *in vitro* the interactions between actin and the actin-binding proteins (ABPs) and to determine how these interactions are affected by the actinopathy-causing mutation. To achieve this, it is essential to accurately describe the actin–ABP interaction dynamics by defining the actin under investigation by both biochemical tools (Greve & Manstein, 2025; Greve et al., 2022) and *in vitro* mechanics.

The residue E334 is located at the hinge region between subdomains 1 and 3 of the actin monomer (Fig. A1 in the Appendix). Specifically, residues 327–334 interact with the cardiomyopathy (CM) loop of myosin (Von Der Ecken et al., 2016), which is a critical region for actin–myosin interactions (Parker et al., 2020). At this location, the actin protomer interacts with a core-contact triad of a positively charged patch of the loop, a feature present in all class II myosins (Greve et al., 2024). A recent study provided evidence for the impact of the mutation on the interaction kinetics of actin with some of its key binding partners, such as cofilin and non-muscle myosins. While the mutant protein was efficiently integrated into the cytoskeletal actin network, its interaction with human class II and V myosin motors showed significant reductions in sliding velocity and actin affinity, potentially leading to impaired cellular functions (Greve et al., 2024).

So far, the effects of the E334Q mutation on the interaction dynamics of cytoskeletal $\gamma$-actin with myosin have been studied only under unloaded conditions with *in vitro* motility assay (IVMA) and transient–kinetic approaches (Greve et al., 2024). However, all motility processes driven by interactions of non-muscle myosin II with $\gamma$-actin filaments such as cell shape maintenance and plasticity, cytokinesis and cell migration have to overcome the load exerted by the intracellular cytoskeleton or the extracellular support. Therefore, analysis of the effects of the E334Q mutation on the actin–myosin interaction dynamics must be conducted under controlled loading conditions. To this end, the dual laser optical tweezers (DLOT) technology has been applied to mechanical measurements on a nanomachine (Pertici et al., 2018) made by an array of myosin motors interacting with a single $\gamma$-actin filament, either wild-type (WT) or carrying the E334Q mutation. The actin filament is attached with the correct polarity to a bead trapped in the focus of the DLOT, which serves as force transducer, and brought to interact with the myosin array extending from the functionalized surface of

a micropipette carried by a three-way nanopositioner, which serves as a movement transducer/actuator. Skeletal muscle myosin is used as sensor myosin, because non-muscle myosin is too slow for reliable DLOT measurements. Using this approach, we demonstrate that, in the presence of 2 mM ATP, all the relevant parameters of the actin–myosin interaction dynamics (maximum shortening velocity, steady isometric force, maximum power and rate of force development) are depressed by the mutation. The force produced by individual myosin–actin interactions, estimated by interpreting the isometric force fluctuations with a stochastic model, is reduced to $\frac{1}{2}$ by the mutation, accounting for the reduction of the ensemble force. Notably, the ability to move under loaded conditions is blunted also for heterofilaments polymerized from 50% WT and 50% p.E334Q monomers, a condition that resembles the heterozygous nature of the mutation *in vivo* (Di Donato et al., 2016).

Taken together, the results indicate that a blunted mechanical performance of the actin–myosin interaction is the molecular basis of the onset of the pathology caused by mutation E334Q of $\gamma$-actin. Moreover, the DLOT technology emerges as an efficient tool for the identification and quantitative definition of actin–myosin interactions altered by NMA-causing mutations in cytoskeletal actin, and the evaluation of small molecule effectors in targeted therapeutic interventions.

## Methods

### Ethical approval

Rabbits (New Zealand white strain, Envigo; weight: 4–5 kg; age: 20–30 weeks) were housed at Centro Stabulazione Animali da Laboratorio (CeSAL, University of Florence) under controlled conditions of temperature and humidity, and kept with free access to food and water prior to use. Two rabbits were used for this study. Animals were first anaesthetized with sodium thiopental (20 mg kg$^{-1}$, i.v. injection into the marginal ear vein) to induce general anaesthesia. Once deep anaesthesia was confirmed, the animals were killed by administering an overdose of sodium thiopental (50 mg kg$^{-1}$) in the marginal ear vein, in compliance with the guidelines of the institution's animal welfare committee [Organismo Preposto al Benessere Animale (OPBA), University of Florence], the Italian regulation on animal experimentation (Decreto Legislativo 26/2014) and EU directive 2010/63. Ethics approval reference number: 17E9C.N.CLU, authorized by the Ministry of Health (Italy).

The authors confirm that their work complies with the animal ethics checklist and with *The Journal of Physiology* policies regarding animal experiments.

## Preparation of proteins

Skeletal muscle myosin was obtained from psoas muscle of two adult male rabbits. Heavy MeroMyosin (HMM) fragments of myosin were purified from rabbit psoas muscle as already reported (Pertici et al., 2018, 2020).

Recombinant human cytoskeletal γ-actin and p.E334Q mutant protein were produced as actin–thymosin β4-His$_6$ fusion proteins using the baculovirus/*Spodoptera frugiperda* insect cell expression system and purified as previously described (Greve et al., 2024).

G-actin was polymerized for 3 h at room temperature by the addition of 2 mM MgCl$_2$ and 100 mM KCl and then incubated overnight at 4°C with an excess of phalloidin-tetramethyl rhodamine iso-thiocyanate (TRITC, Merck, Darmstadt, Germany). TRITC–phalloidin-labelled WT and mutant actin filaments were used to generate bead-tailed actin (BTA). For this purpose, a single phalloidin–rhodamine-labelled actin filament was bound to a functionalized 3 μm polystyrene bead (Kisker Biotech, Steinfurt, Germany) coated with gelsolin (plasma porcine, Hypermol, Bielefeld, Germany) or Ca$^{2+}$-insensitive gelsolin fragment TL40 (Hypermol), using established protocols (Pertici et al., 2020).

## Analysing the mechanical performance of the actin–myosin motile system via a synthetic nanomachine

The ability of the recombinant WT and p.E334Q γ-actin to function as a track for myosin-driven cell motility was analysed using a synthetic nanomachine originally devised for reproducing the emergent properties of muscle myosin working in ensemble (Pertici et al., 2018, 2020).

The choice of myosin II from rabbit skeletal muscle as 'sensor' myosin was dictated by the consideration that the limits intrinsic to the DLOT operated in force clamp (accuracy and reproducibility of the position, thermal noise of the piezo-manipulator and of all the optical components) underpin a noise around 20–30 nm s$^{-1}$, and this makes the measurement of the actin sliding velocity powered by the very slow non-muscle myosin II unreliable. Instead, the reliability of the DLOT technology to measure *in vitro* the mechanical performance of the nanomachine made by an array of skeletal muscle myosin II interacting with α-actin filament is well established (Pertici et al., 2018, 2020, 2021). Under this condition, the nanomachine can be considered a standard reference system for its application to the definition of functionality of other actins.

The construction of the synthetic nanomachine, as well as the detailed mechanical protocols for performing experiments in the absence and the presence of nucleotide have been described in detail previously (Pertici et al.,

2018). Briefly, the two-channel flow chamber hosting the nanomachine consisted of an upper compartment for BTA introduction and a lower compartment with the support for the motors in which the HMM fragment were introduced. The motor support was a glass pipette heat-pulled to a diameter of ∼3 μm and functionalized with 1% nitrocellulose. The support was moved together with the flow chamber by the three-way piezoelectric nanopositioner (nano-PDQ375, Mad City Lab, Madison, WI, USA), acting as a displacement transducer in the range 0–75,000 nm (resolution 1.1 nm). The nanopositioner was carried by a micropositioner (Mad City Lab) for the larger movements necessary to bring the motor ensemble to interact with a BTA trapped into the focus of a DLOT apparatus (Smith et al., 2003), acting as a force transducer. With laser power on the sample of 100 mW, the dynamic range of the force transducer was 0–250 pN (resolution ∼0.5 pN) (Bianco et al., 2011).

The correct orientation of the γ-actin filament on the bead was achieved by exploiting the BTA protocol (Suzuki et al., 1996). The BTA is generated by covalently binding the capping protein gelsolin or its Ca$^{2+}$-insensitive construct TL40 (Pertici et al., 2020) to the polystyrene bead, which ensures a reproducible orientation of the actin filament due to the preference of gelsolin for its barbed end.

The nanomachine setup can be operated in position clamp (PC), achieved by using as feedback signal the position of the nanopositioner carrying the motor array to reproduce the isometric contraction, or in force clamp (FC), by using as feedback signal the position of the bead in the laser trap to reproduce isotonic contraction and measure the sliding velocity as function of the force (*F–V* relation) and the maximum power output (Pertici et al., 2018, 2021). The system has been recently implemented with the possibility to operate in length clamp (LC), achieved by using as feedback signal the distance between the position of the bead and that of the myosin array support (Buonfiglio et al., 2024). The LC mode allows the elimination of sliding between the actin filament and the motor array caused by any force change in the presence of the large trap compliance (∼4 nm pN$^{-1}$) in series to the motile system. In LC, any movement of the bead is counteracted by that of the nanopositioner and the series compliance is reduced to an effective value of 0.2 nm pN$^{-1}$. In this way, the filament sliding produced by a force-generating interaction is much less than 1 nm and thus does not reflect in a significant change in strain of the other attached motors. In this condition, motors behave almost as independent force generators and the rate of development of the steady isometric force in 2 mM ATP is attributable to the kinetics of motor attachment and detachment.

The composition of the buffers used for both the rigor and the active experiments is reported in Pertici et al.

(2018, 2020). Methylcellulose at 0.5% was added to the final buffer in order to inhibit lateral diffusion of the actin filament and maintain the acto-myosin interaction even at low-force isotonic contractions (Pertici et al., 2018; Uyeda et al., 1990).

The TRITC–phalloidin fluorescence of the actin filament was excited with an M503F2 fibre-coupled LED (530 nm, 6.8 mW, Thorlabs, Newton, NJ, USA) and recorded using a sCMOS camera (Kuro 1200B, Teledyne Princeton Instruments, Trenton, NJ, USA). Custom-written LabVIEW algorithms were used for signal generation and data acquisition.

All the experiments were performed at room temperature (24°C).

## Mechanical protocols

**Rupture events in rigor conditions.** To estimate the number of myosin motors available for actin interaction, the myosin ensemble, disposed on the nitrocellulose-coated lateral surface of the micropipette, was first brought into contact with the BTA in ATP-free solution to form the rigor bonds with the actin filament. In PC, after an initial displacement of 1–2 μm in the z direction (orthogonal to the actin–myosin interface, x), the micropipette was pulled along the x-axis with a constant velocity of 50 nm s$^{-1}$, leading to the break of the rigor bonds one at a time. This procedure allows the first bonded HMM to undergo a pulling force higher than the axial component of the force shared among the other motors (Buonfiglio et al., 2024). The low pulling velocity ensures that movement of the nanopositioner during the force drop is negligible, thus allowing each single rupture event to be resolved (Buonfiglio et al., 2024; Nishizaka et al., 2000; Pertici et al., 2018). The force component along the x-axis ($F_x$) was measured and the bead displacement along the x-axis was calculated knowing the trap stiffness (0.25 pN nm$^{-1}$).

Rupture events were identified as force drops > 2 pN, complete in < 50 ms (Pertici et al., 2018). This definition takes into account the noise on the force trace (typically ∼2 pN), setting the minimum detectable amplitude of the force drop, and the condition that at the imposed nanopositioner pulling velocity (50 nm s$^{-1}$) a time > 50 ms (corresponding to an imposed displacement > ∼2.5 nm) could imply not detachment, but reorientation of the motor at the interface with the substrate. In addition, the rise in force following the drop caused by a rupture event must occur in more than 50 ms not to be classified as noise fluctuation. In fact, in < 50 ms the movement of the nanopositioner would be < 2.5 nm, not enough to load the following actin–myosin bond and induce its rupture.

The amplitudes of the force drops are directly related to the distance between consecutive myosin motors randomly distributed on the pipette lateral surface (Pertici et al., 2018).

**Isometric and isotonic contractions in 2 mM ATP.** In experiments with the active nanomachine, a solution containing 2 mM ATP was continuously flowed through the flow chamber at a constant velocity of 3 μL min$^{-1}$. The nanopositioner was moved along the z-axis to bring the motor ensemble towards the laser-trapped BTA. Following formation of the actin–myosin interface, the ensemble of motors started to develop force in PC.

For determination of the force–velocity ($F$–$V$) relation, after the maximum steady force $F_0$ was developed, the control was switched to FC for the imposition of different loads, to which the myosin ensemble responded with sliding of the actin filament in the direction of shortening at constant velocity and inversely proportional to the force (Pertici et al., 2018).

$F$–$V$ data were grouped in classes of 2–3 pN force and averaged. The relevant parameters of the interaction were obtained by fitting the bin averages with Hill's hyperbolic equation (Hill, 1938). For each condition, the choice of the $F$ classes was related to the number of force steps imposed from $F_0$ during the mechanical protocol. In this way, we avoid high-load data points contributing to the fit more than the low-load data. In fact, at low loads ($F \leq 0.3F_0$), due to decreased duty ratio and the limited size of the myosin array, the probability that at least one motor is attached at any time (the condition that ensures continuous interaction) becomes progressively lower with decreasing force (Pertici et al., 2018, 2021). For this reason, the low-load region of the $F$–$V$ curve is typically underrepresented in terms of data points with respect to the region of the high loads (see also the $F$–$V$ relations in Fig. 2C).

For the estimate of the rate of force development under truly isometric conditions, after attainment of $F_0$ the control was rapidly switched to LC and a rapid shortening of ∼250 nm was imposed to make the force drop to zero. Force redevelopment in LC occurred with a rate ($k_t$) not influenced by the trap compliance and thus is the expression of the kinetics responsible for the transition to the steady force (Buonfiglio et al., 2024). During force redevelopment in LC, the nanopositioner moved away from the trap by an amount necessary to counteract the movement of the bead ($d$, ∼50 nm).

## *In vitro* motility assay

IVMA, measuring the sliding velocity ($V_f$) of the actin filaments under investigation on a bed of psoas HMM, was performed at room temperature (24°C) and at ∼60 mM ionic strength as previously described (Kron & Spudich, 1986; Pertici et al., 2021). The composition of the experimental solution for the IVMA was: 2 mM

Mg-ATP, 25 mM imidazole pH 7.4, 25 mM KCl, 1 mM EGTA, 4 mM MgCl$_2$, 10 mM DTT, 20 µg mL$^{-1}$ catalase, 100 µg mL$^{-1}$ glucose oxidase, 5 mg mL$^{-1}$ glucose and 0.5% methylcellulose (w/v, 400 cP, Merck). The concentration of rabbit psoas HMM deposited over the nitrocellulose-coated coverslip was the same selected for the nanomachine experiments in 2 mM ATP (0.2 mg mL$^{-1}$). After HMM deposition, non-fluorescent actin was added to block the dead (rigor-like) HMM (Pertici et al. 2018).

For fluorescence signal acquisition, the flow chambers were mounted on the DLOT apparatus. Sequences of images were acquired at a rate of 2–4 frames s$^{-1}$. The analysis of $V_f$ was performed by exploiting the centroid movement method, following the change in the position of the centre of mass of the actin filament under inspection in consecutive images, with a custom-built program developed with LabVIEW software (National Instruments, Austin, TX, USA). The sliding velocity $V_f$ was obtained from the average of the velocity distributions of $n$ areas with dimensions of 70 × 70 µm and a similar number (5–10) of filaments ($n$ = 46 for γ-WT, 32 for p.E334Q and 44 for heterofilaments).

### Modelling of the mechanical output of the nanomachine

The stochastic model described in Buonfiglio et al. (2024) was adopted to analyse the time series of the transient and steady-state force exerted by the ensemble of motors during the cyclical acto-myosin interactions. Implementation of the nanomachine with the LC mode developed in Buonfiglio et al. (2024) allows minimization of the trap compliance and recovery of the condition of the attached myosin motors to work as independent force generators during the transition to the steady state and the force fluctuation at steady state. For this reason, we considered the ensemble of $N$ motors able to interact with actin defined by the rupture events in rigor as independent molecular motors in isometric contractions.

A detailed analysis of the stochastic model and of the optimization procedure for the parameter estimation can be found in the Methods section in Buonfiglio et al. (2024). In the Appendix we provide a description of the stochastic model and a brief overview of the theoretical approach. The information stemming from the force records allows us to estimate the duty ratio $r$, the single motor force $f_0$ and the effective flux of motors through the cycle per unit time $\varphi$. The fitting procedure was applied to the experimental data collected with the nanomachine powered by either WT or p.E334Q γ-actin filament, in interaction with an ensemble of HMMs purified from rabbit psoas. See Fig. 4 for a graphical representation of the results of the optimization procedure.

### Statistical analysis

Data are presented as mean ± SD. The number of individual experiments $n$ is reported in figure legends and/or throughout the text and tables.

One-way ANOVA or two-way ANOVA with Tukey *post hoc* test was used for statistical testing as reported in each table legend. JASP 0.95 (JASP Team, 2025) and Origin Pro 2025 (OriginLab, Northampton, MA, USA) were used for statistical testing. The null hypothesis was rejected for $P \leq 0.05$. Exact $P$ values are reported in the text, except when $P$ is less than 0.001 ($P < 0.001$).

For the best fit parameters and their associated uncertainties obtained from Hill's hyperbolic fit to the experimental data, variations from the control were considered significant when the parameter value deviates from the corresponding γ-WT value by more than two SD (as derived from the fitting procedure).

## Results

### The E334Q mutation reduces the affinity of the acto-myosin interaction as measured by the number of rupture events in rigor

Myosin II from rabbit skeletal muscle (psoas) was selected as the 'sensor' myosin for the application of the nanomachine to determine the effect of the E334Q mutation on the performance of the γ-actin–myosin system (see Methods). First, the number of myosin molecules, from those on the support, available for the interaction with the actin filament must be estimated. For this, the system is operated in PC mode (see Methods), in which the position of the nanomanipulator carrying the myosin ensemble is used as feedback signal (Buonfiglio et al., 2024; Pertici et al., 2018). The myosin ensemble, deposited on the functionalized support, is brought into contact with the actin filament in the absence of ATP, to form actin–myosin rigor bonds (Fig. 1*A*, panel 1; Nishizaka et al., 2000; Pertici et al., 2018). Once the rigor interaction is established, the pipette is first displaced from the actin filament in the direction (z) orthogonal to the actin–myosin interface (x), and then pulled away along the x-axis with a constant velocity of 50 nm s$^{-1}$ (panel 2). Since the force is applied along a diagonal, the rigor bonds are broken one at a time, preventing the HMM from rebinding after its detachment from actin. These rupture events can be identified as a rapid force decrease following a build-up of force (Fig. 1*B–D*), the last of which corresponding to the complete detachment of the actin filament. The rupture event concerns both heads of each HMM, considering that the two heads attach to two consecutive actin monomers of the same actin filament strand and thus are separated by only ∼5.5 nm (Reconditi et al., 2003), and that the trap compliance is 4 nm pN$^{-1}$

in the x–y plane and much larger along the z-axis. The assumption of two heads per rupture was solidified by the model simulation of the nanomachine performance in the presence of physiological [ATP], predicting a number of available motors twice the number of rupture events in rigor (Pertici et al., 2018).

The protocol has a specific application in the present study, as it allows us to establish whether the reduced affinity between p.E334Q $\gamma$-actin filaments and NM IIA reported in a previous work (Greve et al.,

2024) is reproducible with the 'sensor' myosin under the experimental conditions of the nanomachine. In the experiments performed with WT $\gamma$-actin (Fig. 1*B*), with the standard concentration of psoas HMM (Pertici et al., 2018) of 0.1 mg mL$^{-1}$, we found 7.9 $\pm$ 1.6 rupture events, indicative of eight HMM molecules forming rigor bonds with the actin filaments. Saturation of the rupture events occurred at [HMM] = 0.2 mg mL$^{-1}$, at which the number of rupture events was 10.1 $\pm$ 2.5 (Table 1, Fig. 1*B*).

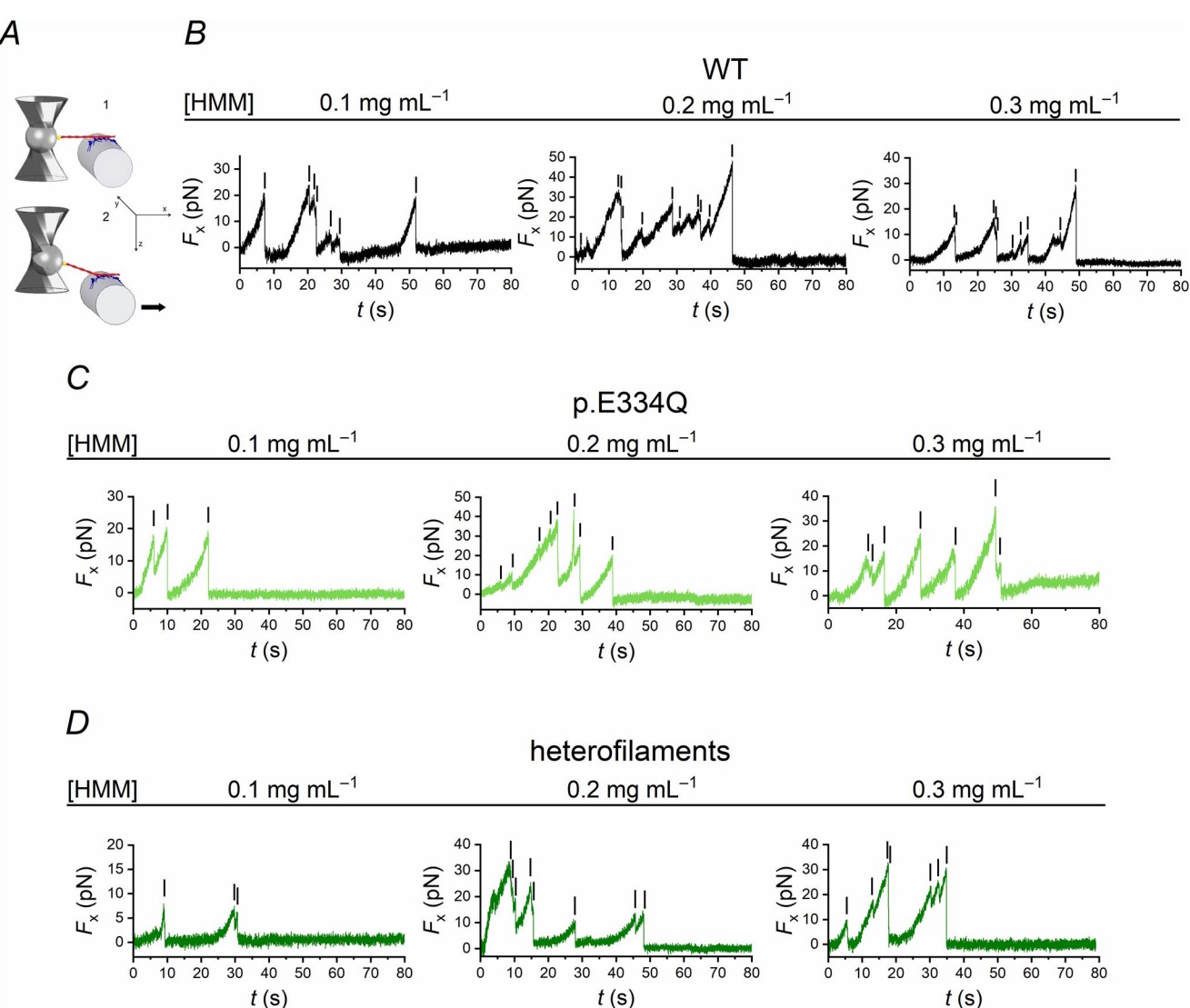

**Figure 1. Determination of the number of HMMs available for the interaction with WT, p.E334Q $\gamma$-actin and heterofilaments from rupture events in the absence of ATP**

*A*, scheme showing the experimental setup: HMM molecules (blue) are immobilized on a glass pipette and brought into contact with a phalloidin-stabilized, TRITC-labelled actin filament (red), which is linked to a laser-trapped bead (grey) via the actin binding domain of gelsolin (yellow). After formation of the acto-myosin rigor bonds (shown in 1), the pipette is moved in a direction (z) orthogonal to the actin–myosin interface and then moved in the direction (x) marked by the arrow to break the rigor bonds one at a time (shown in 2). The scheme is modified from Pertici et al. (2018). *B–D*, representative traces for WT (black, *B*), p.E334Q $\gamma$-actin (light green, *C*) and heterofilaments (dark green, *D*) performed at the indicated HMM concentrations. Ruptures of individual rigor bonds are marked by the black vertical lines in the x-component of the force ($F_x$)–time plot.

**Table 1. Rupture events in ATP-free solution**

| | WT | | |
|---|---|---|---|
| [HMM] (mg mL⁻¹) | 0.1 | 0.2 | 0.3 |
| Rupture events | $7.9 \pm 1.6$ | $10.1 \pm 2.5$ | $10.2 \pm 1.9$ |
| *n* | 22 | 31 | 13 |
| | **p.E334Q** | | |
| [HMM] (mg mL⁻¹) | 0.1 | 0.2 | 0.3 |
| Rupture events | $2.9 \pm 1.5$ (-63%)*** | $6.6 \pm 2.5$ (-35%)*** | $7.0 \pm 2.4$ (-31%)* |
| *P* | < 0.001 | < 0.001 | 0.0241 |
| *n* | 12 | 29 | 11 |
| | **Heterofilaments (1:1)** | | |
| [HMM] (mg mL⁻¹) | 0.1 | 0.2 | 0.3 |
| Rupture events | $4.0 \pm 2.0$ (-49%)** | $7.7 \pm 2.8$ (-24%)* | $7.5 \pm 2.2$ (-26%)* |
| *P* | 0.00163 | 0.0232 | 0.0494 |
| *n* | 10 | 16 | 16 |

Number of observed rupture events at different myosin concentrations are given as the mean $\pm$ SD of *n* individual experiments. Values in parentheses are percentage variations with respect to γ-WT at the same HMM concentration. Statistical significance was assessed with two-way ANOVA followed by Tukey's *post hoc* test (*$P < 0.05$; **$P < 0.01$; ***$P < 0.001$).

Performing the experiment with p.E334Q γ-actin filaments (Fig. 1*C*), we measured $2.9 \pm 1.5$ rupture events with [HMM] = 0.1 mg mL⁻¹ (Table 1). The number of ruptures increased with further increase of [HMM] attaining, at 0.2 mg mL⁻¹, a value approximately twofold higher ($6.6 \pm 2.5$). A further increase of [HMM] to 0.3 mg mL⁻¹ did not significantly increase the number of rupture events ($7.0 \pm 2.4$) (Table 1). These results are indicative of a reduced affinity of the p.E334Q γ-actin for skeletal muscle HMM in ATP-free solution, in agreement with the reduced affinity of the p.E334Q γ-actin for non-muscle HMM measured using transient-kinetic approaches under rigor conditions (Greve et al., 2024). In turn, the mutation-dependent loss of affinity of the mutant actin for either myosin solidifies the reliability of the selection of skeletal muscle myosin as the 'sensor' myosin to establish the effect of the E334Q mutation on the γ-actin–myosin mechanics.

The same experiment was repeated using heterofilaments, in which monomers of WT and p.E334Q γ-actin were co-polymerized in a 1:1 ratio to simulate the heterozygous nature of the disease in patients (Fig. 1*D*). At [HMM] of 0.1 mg mL⁻¹, the number of ruptures was $4.0 \pm 2.0$ (Table 1), still significantly reduced with respect to WT γ-actin filaments. The number of rupture events reached $7.7 \pm 2.8$ at an HMM concentration of 0.2 mg mL⁻¹ (Table 1).

The HMM concentration of 0.2 mg mL⁻¹ was chosen for all the experiments in 2 mᴍ ATP. Under this condition, each head of the skeletal muscle myosin II dimer works independently (Cooke & Franks, 1980; Pertici et al., 2018), and thus the number of motors available for actin interaction (*N*) is twice the number of the rupture events, that is $20 \pm 1$ for WT γ-actin and $14 \pm 1$ for p.E334Q γ-actin.

**p.E334Q γ-actin reduces the force and the power generated by the interaction with the myosin II ensemble**

The effect of the E334Q mutation on the force and shortening produced during the interaction of the γ-actin filament with the myosin ensemble in 2 mᴍ ATP is shown in Fig. 2. As the myosin ensemble is brought into contact with the actin filament by moving the nanopositioner in PC mode (Fig. 2*A* and *B*) and the actin–myosin interaction is established, the motors start to produce force (*F*) (phase 1) that eventually reaches a steady value $F_0$, which measures the maximum force capability of the myosin ensemble. At $F_0$, the system is switched to FC

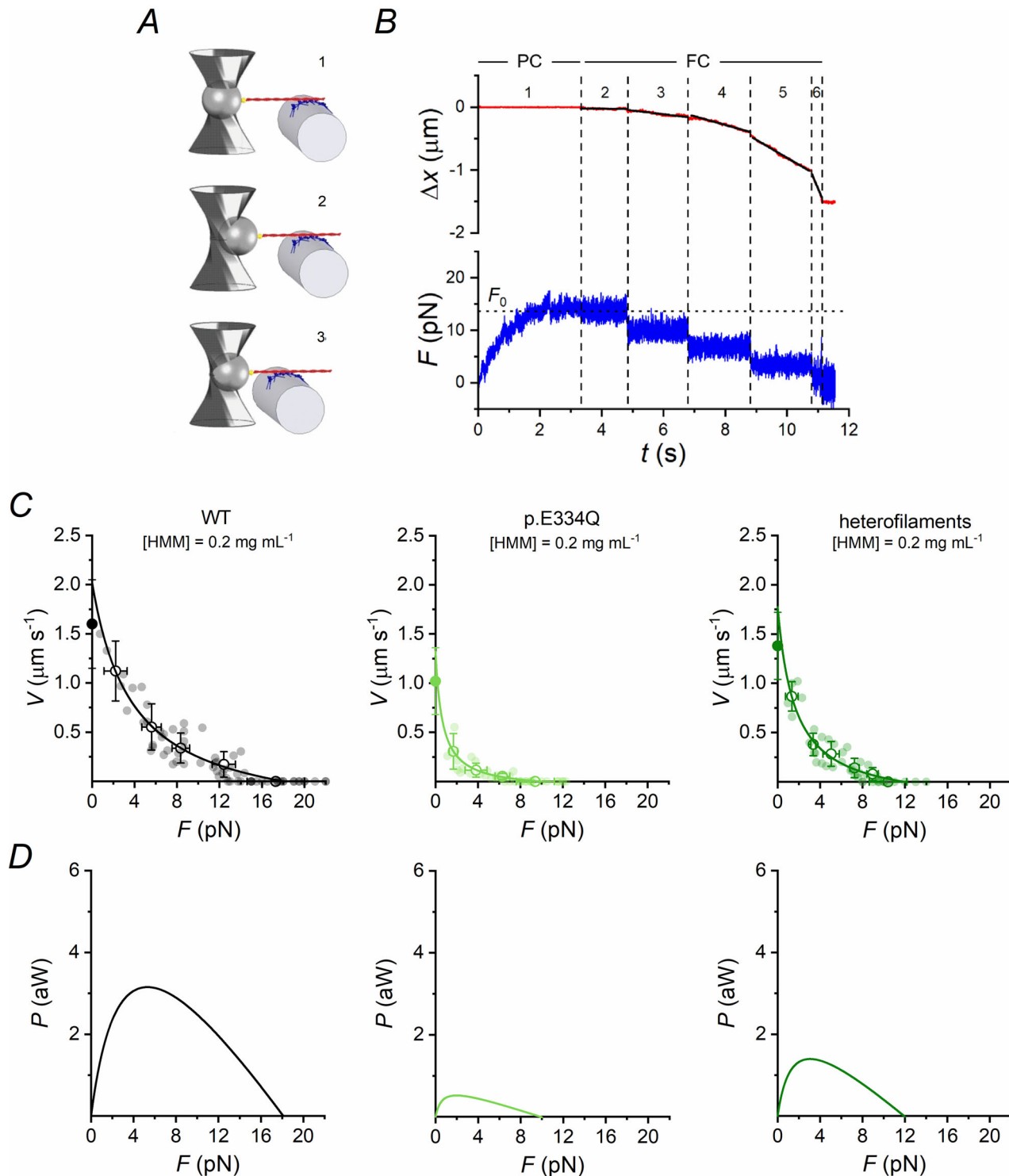

**Figure 2. Interaction of WT/p.E334Q $\gamma$-actin filaments and heterofilaments with a skeletal myosin ensemble under load analysed with the synthetic nanomachine in 2 mM ATP**

*A*, scheme representing three snapshots during the phases of the interaction between the actin filament and the motors, numbered as in *B* [1, before force development; 2, during maximum isometric force ($F_0$) after the switch between PC and FC; 3, during isotonic shortening after the first force drop]. Reproduced from Pertici et al. (2018). *B*, representative trace showing a recording of the reciprocal sliding ($\Delta x$, upper trace, red) and force (*F*, lower trace, blue) during an interaction between the ensemble of HMM and WT $\gamma$-actin. Phase 1, after the formation of the acto-myosin bonds, the force rises under PC to $F_0$ ($\sim$14 pN); phases 2–6, shortening response to a staircase of

stepwise reductions in force (3 pN) separated by 2 s under FC. Black lines superimposed on the sliding trace during phases 2–6 are the linear fits from which the sliding velocity $V$ is calculated. C, force–velocity (F–V) relations from experiments performed with WT γ-actin (black, $n = 25$), p.E334Q γ-actin (light green, $n = 19$) and heterofilaments (dark green, $n = 13$). For each condition, individual data points (small, filled dots) were grouped in classes of 2–3 pN force and averaged (empty dots, mean ± SD). The continuous line in each plot represents Hill's hyperbolic fit to the average value of each class. In each plot, the filled dot on the y-axis is the experimentally determined sliding velocity of the respective filaments on psoas HMM in the unloaded IVMA ($V_f$). D, P–F relations ($P = F \times V$), calculated from the corresponding F–V relations and identified with the same colour code. The parameters extracted from the relations in C and D and their statistical outcomes are provided in Table 2.

**Table 2. Data derived from nanomachine experiments performed in the presence of 2 mM ATP**

| | WT | p.E334Q | Heterofilaments (1:1) |
|---|---|---|---|
| $V_f$ (µm s$^{-1}$) | 1.60 ± 0.45 | 1.02 ± 0.34 (−36%) $P < 0.001$*** | 1.38 ± 0.39 (−14%) $P = 0.0285$* |
| $V_0$ (µm s$^{-1}$) | 2.02 ± 0.15 | 1.32 ± 0.71 (−35%) | 1.78 ± 0.37 (−12%) |
| $F_0{}^*$ (pN) | 18.1 ± 1.0 | 9.9 ± 1.1 (−45%)* | 11.9 ± 1.7 (−34%)* |
| $P_{max}$ (aW) | 3.15 ± 0.44 | 0.52 ± 0.20 (−83%)* | 1.39 ± 0.34 (−56%)* |
| $a/F_0{}^*$ | 0.21 ± 0.05 | 0.07 ± 0.05 (−67%)* | 0.14 ± 0.07 (−33%) |
| $k_t$ (s$^{-1}$) | 10.5 ± 2.4 | 7.3 ± 2.3 (−30%) $P = 0.00641$** | 9.1 ± 1.8 (−13%) $P = 0.317$ |

$V_f$, velocity of actin sliding in IVMA; $V_0$, maximum sliding velocity estimated from the y-intercept of the hyperbolic fit; $F_0{}^*$, mean isometric force, estimated from the fit x-intercept; $P_{max}$, maximum power; $a/F_0{}^*$, curvature; $k_t$, rate of force redevelopment estimated from LC experiments. $V_f$ and $k_t$ are expressed as mean ± SD over independent experiments, while $V_0$, $F_0{}^*$, $P_{max}$ and $a/F_0{}^*$ are extracted from the best fit, together with their associated uncertainties as provided by the fitting software. The $n$ of the different experiments and conditions is reported in the corresponding paragraphs and figure legends. Percentage variations with respect to γ-WT are shown in parentheses. For the best fit parameters, differences are marked with an asterisk (*) when the parameter value deviates from the corresponding γ-WT value by more than 2 SD (as derived from the fitting procedure). One-way ANOVA followed by Tukey's *post hoc* test was employed for statistical testing of $V_f$ and $k_t$ (*$P < 0.05$; **$P < 0.01$; ***$P < 0.001$).

mode (see Methods), in which the feedback signal is the displacement of the bead from the focus of the lasers (phase 2). A staircase of stepwise reductions in force is imposed (phases 3–6, 3 pN steps separated by 2 s). The myosin ensemble responds to the force drop inducing sliding of the actin filament in the direction of shortening at constant velocity ($V$) that is inversely proportional to the force according to the F–V relation (Hill, 1938). In Fig. 2C the F–V relations for WT γ-actin (black), p.E334Q γ-actin (light green) and heterofilaments (dark green) are shown. Individual data points (small filled circles) were grouped in classes of 2–3 pN force and averaged (open circles). Open circles were fit by the hyperbolic equation (Hill, 1938) (continuous lines):

$$(F + a)\,(V + b) = \left(F_0{}^* + a\right) b,$$

where $a$ and $b$ are the distances from the y-axis and the x-axis of the vertical and horizontal asymptote, respectively, and $F_0{}^*$ is the intercept of the hyperbole on the x-axis, which may be slightly higher than $F_0$ (Cecchi et al., 1978; Edman et al., 1976). The distance $a$ normalized

for $F_0{}^*$ ($a/F_0{}^*$) is an effective estimate of the radius of curvature of the F–V relation.

$F_0{}^*$ was 18.1 ± 1.0 pN with WT γ-actin and decreased by 45% with p.E334Q γ-actin (9.9 ± 1.1 pN) (Table 2). The ratio $a/F_0{}^*$ too was reduced in the presence of the mutation (from 0.21 ± 0.05 with WT γ-actin to 0.07 ± 0.05 with p.E334Q γ-actin). The maximum shortening velocity ($V_0$, the intercept of the hyperbole on the y-axis) decreased from 2.02 ± 0.15 to 1.32 ± 0.71 µm s$^{-1}$, a reduction of 35% but that is not significant (i.e. within 2 SD difference from the γ-WT value, see Methods), also because of the large error associated with force measurements for low fractions of the isometric force in the mutant. In fact, in the mutant the low force values are < 1 pN, the threshold value for the force resolution of the nanomachine. In this respect, it must be noted that $V_f$, the velocity of sliding in *in vitro* motility experiments conducted before the mechanical experiments to test the functionality of the nanomachine components, was also reduced by ∼40% with the mutant actin and in this case the difference was significant ($P < 0.001$). The F–V relation obtained with

the heterofilaments shows that the depressing effects of the mutation on $F_0^*$ were only partially attenuated but still highly significant: in this case $F_0^*$ was $11.9 \pm 1.7$ pN (Fig. 2*C*, Table 2), 34% smaller than WT $\gamma$-actin.

For each *F–V* point the steady power (*P*) can be calculated by the product $F \times V$. The *P–F* relations, derived from each hyperbolic fit, allow us to estimate the maximum power ($P_{\max}$) (Fig. 2*D*). The decreases in both $F_0^*$ (−45%) and $a/F_0^*$ (−67%), found with p.E334Q $\gamma$-actin with respect to WT $\gamma$-actin, underpin an even stronger reduction in $P_{\max}$ (from $3.15 \pm 0.44$ to $0.52 \pm 0.20$ aW, −83%, Table 2). The incomplete recovery of both $F_0^*$ (−34%) and $a/F_0^*$ (−33%) with the heterofilament implies the maintenance of a highly significant depression of $P_{\max}$ ($1.39 \pm 0.34$ aW, −56%, Table 2).

### p.E334Q $\gamma$-actin reduces the rate of force development in isometric conditions

Among the emergent properties of the interaction of a myosin II ensemble with the actin filament, the rate of the rise of force to the maximum steady value provides information on the kinetics of myosin motor attachment and detachment, provided that the force rise occurs in truly isometric conditions. This is not the case for the force rise in PC shown in Fig. 2*B*, which takes about 2 s, as in this case the trap compliance ($\sim$4 nm pN$^{-1}$) implies movement of the bead away from the laser focus during force rise with an overall shortening of $\sim$50 nm. With the nanomachine in LC mode (see Methods and Buonfiglio et al., 2024), the feedback signal is the change in the distance between the position of the bead and the myosin array; in this case, during the rise of force, actin filament sliding is prevented so that motors behave almost as independent force generators and the rate of force rise to the maximum steady value ($k_t$) is attributable only to motor attachment–detachment kinetics.

To estimate $k_t$, at the plateau of isometric force ($F_0$) the control is switched from PC to LC and a ramp-faced shortening (of $\sim$250 nm complete within 125 ms) is imposed to drop the force to zero (Fig. 3). There is a minimum time for which the force remains at zero after the end of the ramp, and then the force redevelops up to the original $F_0$ value, as expected if the number of available motors remains constant (Pertici et al., 2018). The time course of the force redevelopment, one order of magnitude shorter than that in PC in Fig. 2*B*, can be fitted with a single exponential equation to estimate the time constant $\tau$ (the time required to reach $0.63F_0$). The value of $\tau$ was $95 \pm 22$ ms (mean $\pm$ SD) for WT $\gamma$-actin ($n = 21$), $137 \pm 43$ ms for E334Q $\gamma$-actin ($n = 7$) and $110 \pm 22$ ms for the heterofilament ($n = 9$). The rate of

force development $k_t$ ( $= 1/\tau$) was therefore $10.5 \pm 2.4$ s$^{-1}$ for WT $\gamma$-actin and decreased by 30% for E334Q $\gamma$-actin ($7.3 \pm 2.3$ s$^{-1}$, $P = 0.00641$) (Table 2). The value of $k_t$ for the heterofilament was also reduced ($9.1 \pm 1.8$ s$^{-1}$, −13%) but not by a statistically significant amount ($P = 0.317$).

### E334Q mutation reduces the force exerted by a single actin–myosin interaction without affecting the duty ratio

The rate of transition to the isometric force $F_0$ and the force fluctuations superimposed on $F_0$ in LC can feed a stochastic model, which provides a self-consistent estimate of the implicit mechano-kinetic parameters of the interaction between the actin filament and the myosin ensemble. This analysis has been described in detail previously (Buonfiglio et al., 2024), where it was used to compare the force and the kinetics of individual actin–myosin interactions with the nanomachine powered by myosin purified from the slow (soleus) and fast (psoas) skeletal muscle of the rabbit. The hidden mechano-kinetic parameters estimated by the model are: $f_0$, the force of a single motor interaction; $r$, the duty ratio (the fraction of time the motor stays attached with respect to the total time required to complete the ATP hydrolysis, corresponding to the fraction of motors attached at any time); and $\varphi$, the rate of transition through the attachment/detachment cycle, corresponding to the rate of ATP hydrolysis (Buonfiglio et al., 2024).

The model simulation requires us to know $N$ (number of motors available for actin interaction), which in 2 mM ATP is assumed to be twice the number of rupture events in rigor. For the WT $\gamma$-actin, with $N = 20$, the output of the model gave estimates of $2.48 \pm 0.48$ pN for $f_0$, $0.60 \pm 0.08$ for $r$ and $2.54 \pm 0.18$ s$^{-1}$ for $\varphi$ (Fig. 4 and Table 3). For the p.E334Q $\gamma$-actin, twice the number of rupture events is 14. The output of the model gave estimates of $1.25 \pm 0.26$ pN for $f_0$, $0.77 \pm 0.05$ for $r$ and $1.30 \pm 0.22$ s$^{-1}$ for $\varphi$ (Fig. 4 and Table 3). Thus, the mutation depresses to $\frac{1}{2}$ both $f_0$ and $\varphi$ ($P = 0.00128$ and $< 0.001$, respectively), while it increases $r$ by 28% ($P = 0.00684$). However, considering that the affinity of myosin for actin has been found to depend on the presence of ADP (Greve et al., 2024), in 2 mM ATP the actin–myosin affinity with the mutant actin could have been recovered by a given amount. Assuming as the upper limit a total recovery, $N$ can be taken as 20 at most (the value from the number of rupture events with WT $\gamma$-actin). In this case, the output of the model with the p.E334Q $\gamma$-actin gave $1.02 \pm 0.25$ pN for $f_0$, $0.67 \pm 0.06$ for $r$ and $1.63 \pm 0.13$ s$^{-1}$ for $\varphi$ (Table 3). Thus, also in this extreme case, both $f_0$ and $\varphi$ are significantly decreased by the mutation ($P < 0.001$ for both parameters), while the increase in $r$ is no longer significant ($P = 0.294$).

## Discussion

In this paper, DLOT mechanics are exploited for an unprecedented investigation of the effect of the E334Q γ-actin mutation on the γ-actin-competent dynamics of the submembrane cytoskeleton, which is essential for cellular processes such as cytokinesis, apoptosis and cell locomotion. In this regard, the mechanics of the interaction between a γ-actin filament and an array of myosin motors are analysed under controlled loading conditions according to the nanomachine design implemented by us (Pertici et al., 2018). DLOT quantitative measurements of the mechanical parameters of the interaction of WT and E334Q mutant γ-actin with myosin can be made using the HMM fragment of myosin from rabbit psoas muscle as 'sensor' myosin. The mechano-kinetic properties of

this myosin isoform have been fully characterized in the nanomachine (Buonfiglio et al., 2024; Pertici et al., 2018, 2020). The choice of this isoform over the competent NM II was dictated by the very slow sliding velocity of non-muscle myosin, which would have prevented reliable recording of the force–velocity relation. The validity of the choice of skeletal muscle myosin to test the effect of the E334Q γ-actin mutation is further solidified by the consideration that the core contact triad of the CM loop consists of the same residues (I420, V427 and K429 for NM IIC) for all class II myosins (Greve et al., 2024).

We first evaluated the reliability of the motile system to define the functional differences between the WT and p.E334Q γ-actin using the IVMA. The results showed that the actin sliding velocity ($V_f$) of p.E334Q γ-actin over a

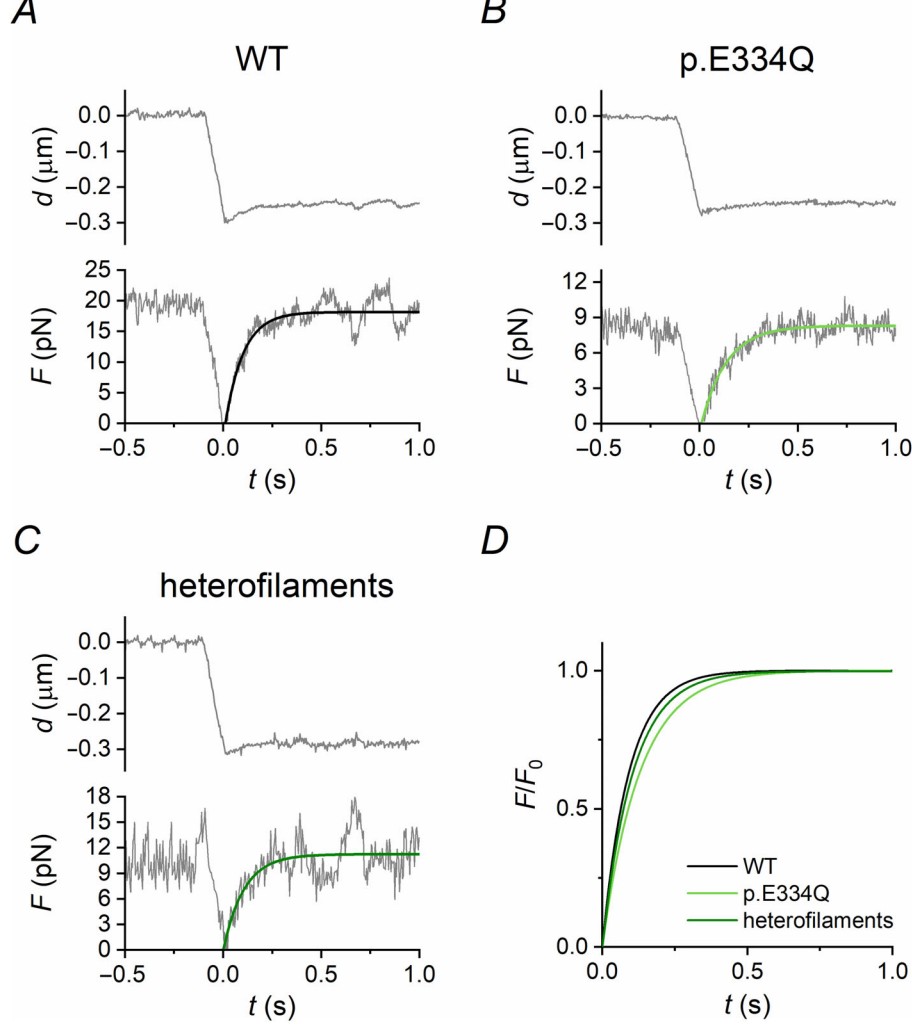

**Figure 3. Representative traces of force redevelopment after rapid shortening in 2 mM ATP**
*A–C*, in response to a 250 nm shortening in LC (upper panel; *d* is the movement of the nanopositioner), force (lower panel) redevelops to the original $F_0$ for WT (*A*), p.E334Q γ-actin filaments (*B*) and heterofilaments (*C*) interacting with the skeletal myosin ensemble. The continuous lines superimposed on the experimental traces are single exponential fits, with the same colour code as in Figs 1 and 2. *D*, exponential fits from *A*, *B* and *C* are normalized each by the respective $F_0$ to better appreciate the time course of force redevelopment.

**Table 3. Relevant mechano-kinetic parameters estimated by the stochastic model**

| | WT | p.E334Q (*N* = 14) | p.E334Q (*N* = 20) |
|---|---|---|---|
| $f_0$ (pN) | 2.48 ± 0.48 | 1.25 ± 0.26 (−50%) *P* = 0.00128** | 1.02 ± 0.25 (−59%) *P* < 0.001*** |
| *r* | 0.60 ± 0.08 | 0.77 ± 0.05 (+28%) *P* = 0.00684** | 0.67 ± 0.06 (+12%) *P* = 0.294 |
| $\varphi$ (s$^{-1}$) | 2.54 ± 0.18 | 1.30 ± 0.22 (−49%) *P* < 0.001*** | 1.63 ± 0.13 (−36%) *P* < 0.001*** |

$f_0$, force per correctly oriented motor; *r*, duty ratio; $\varphi$, rate of transition through the attachment–detachment cycle. Values are mean ± SD from over than four data records for each condition. One-way ANOVA followed by Tukey's *post hoc* test was employed for statistical testing. Values in parentheses are percentage variations with respect to $\gamma$-WT (**P* < 0.05; ***P* < 0.01; ****P* < 0.001).

bed of psoas HMM was 1.6-fold lower than that of WT $\gamma$-actin (Table 2). A significant decrease is also reported for $V_f$ of NM IIA-HMM and mutant actins (Greve et al., 2024). Notably, using the nanomachine assay to measure the number of rupture events in ATP-free conditions (rigor), we obtained a reduced number with p.E334Q $\gamma$-actin at both partial and saturating concentration of psoas HMM, an indication of the reduced affinity between myosin and p.E334Q $\gamma$-actin (Table 1), less pronounced but in agreement with the reduced affinity of p.E334Q $\gamma$-actin for NM IIA-HMM in rigor reported in Greve et al. (2024).

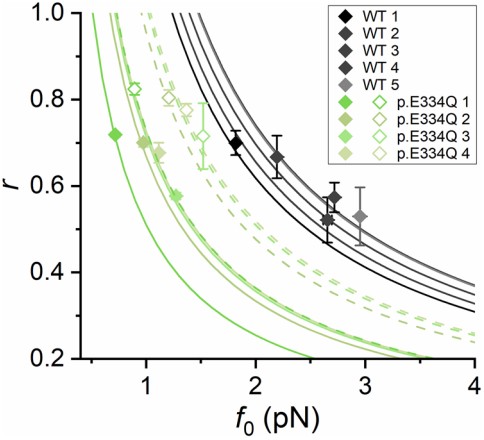

**Figure 4. Estimated motor force $f_0$ and duty ratio *r* by the application of the stochastic model to the experimental data**
Best fit parameters for the arrays of HMM interacting with WT (greyscale diamonds, mean ± SD) and p.E334Q $\gamma$-actin filaments assuming *N* = 20 (light green filled diamonds, mean ± SD) and *N* = 14 (light green empty diamonds, mean ± SD). Different hues correspond to different experimental traces. Mean data are generated by the average of independent realizations of the stochastic fitting procedure. The lines represent the hyperbola set by $F_0$ on which each pair ($f_0$, *r*) is constrained, according to mean field analysis (see the Appendix and Buonfiglio et al., 2024) (greyscale, WT; light green continuous, p.E334Q with *N* = 20; light green dashed, p.E334Q with *N* = 14). The mean values of $f_0$ and *r* and their statistical outcomes are provided in Table 3.

In the presence of physiological [ATP], almost all the relevant mechanical parameters of the interaction of the $\gamma$-actin filament with the myosin array (isometric force, $F_0^*$, maximum shortening velocity, $V_0$, maximum power, $P_{max}$, and curvature of the *F–V* relation, expressed by $a/F_0^*$) are decreased by the mutation (Fig. A2 in the Appendix, Table 2). $V_0$ too is reduced, even if the reduction is not significant, probably due to the scarce description of the *F–V* curve of the mutant actin at low loads. Notably, $V_f$ was ∼20–30% lower than $V_0$, a finding that has been systematically observed comparing the sliding velocity of surface motility assays with that in unidimensional motility systems such as the nanomachine and attributed to non-specific interactions between the actin filaments and the surface in IVMA (Elangovan et al., 2012; Ishijima et al., 1996; Pertici et al., 2018).

Estimates of the hidden parameters of the actin–myosin chemo-mechanical cycle, obtained by fitting force fluctuations in LC with the stochastic model, provide a more detailed description at the molecular level of the mechanical and kinetic properties underpinning the nanomachine performance and its blunting as a consequence of the E334Q $\gamma$-actin mutation. The halving of the ensemble force $F_0^*$ of the nanomachine with the mutant actin is almost totally accounted for by a similar reduction of the single motor force $f_0$. Accordingly, the fraction of attached motors *r* is only marginally affected. On the other hand, $P_{max}$ produced by the nanomachine in the presence of the E334Q $\gamma$-actin mutation decreases to one-fifth of the WT $\gamma$-actin value, a reduction much larger than that of $F_0^*$, which is explained by the contribution of reduced unloaded shortening kinetics accounting for the reduction in $V_f$ and $V_0$. An overall decrease in the attachment–detachment rate constant with the E334Q $\gamma$-actin mutation, which also includes the isometric actin–myosin interactions, is demonstrated by the 30-50% reduction of the kinetic parameters gained with the measurements in LC, namely $k_t$, the rate constant for isometric force development, and $\varphi$, the rate of transition through the cycle.

Our results indicate that the E334Q mutation weakens the ability of $\gamma$-actin filaments to support myosin-based contractility under the full range of physiological loads. This loss of force and power is particularly relevant for related processes of cell motility such as cytokinesis and migration, where $\gamma$-actin networks directly provide the scaffold for myosin II to generate and transmit tension (Maupérin et al., 2025; Vicente-Manzanares et al., 2009). A blunted actin–myosin interaction in the cell cortex is expected to slow down contractile ring closure during cytokinesis and/or impair the force production required for cell migration.

Notably, all the depressant effects of the mutation were attenuated but still significantly present when the p.E334Q $\gamma$-actin monomers were co-polymerized with WT $\gamma$-actin in a 1:1 ratio to simulate the heterozygous nature of the disease in patients (Fig. A2). This condition may be a simplification of the actin filament network composition in patients' cells, as mutant p.E334Q monomers may be subjected to proteasome degradation or other unknown mechanisms regulating the protein level of mutant actin. At present, direct quantitative measurements of the levels of mutant actin in the cell (i.e. using MS) are hindered by the little or no availability of patient-derived cells. However, as previous findings have shown that the E334Q mutation does not affect the folding or stability of the monomer and of the filamentous actin, probably excluding E334Q mutant actin degradation as the primary disease mechanism (Di Donato et al., 2024; Greve et al., 2024), we find it reasonable to assume the 1:1 WT:mutant actin ratio in the patients' cells, at least as an upper bound.

*In vivo*, actin filaments are typically decorated with tropomyosin, a key partner expressed in multiple iso-forms, which modulates actin turnover dynamics and length, controls the association with the filaments of other actin-binding proteins and participates in stress fibre regulation (Manstein et al., 2020). In this study we exploited the possibility offered by our *in vitro* setup to reassemble the contractile system piece-by-piece to focus specifically on the characterization of the actin–myosin mechanical interaction, overcoming the challenge of the complex modulatory contribution of tropomyosin, whose effects could, however, be tested with different degrees of reconstitution. The study of tropomyosin-dependent modulation of the contractile apparatus properties may be relevant in this context, as residue E334 is localized in a region critical for tropomyosin binding (von der Ecken et al., 2016).

This study demonstrates the power of our DLOT technology to compare the performance of the WT and mutant cytoskeletal $\gamma$-actin with a peculiar actin-binding protein, in this case myosin, and identify the loss of basic function underlying cell motility processes such as cytokinesis and migration. However, to link the structural and functional changes responsible for impairment of cell motility to the observed phenotype remains an open challenge for the study of NMAs. Since the most prominent effects of the mutation are evident at the cortical level (Di Donato et al., 2016), the observed defects in brain and neural development should be associated with the impaired interaction between mutant $\gamma$-actin and non-muscle myosin in neuronal cells, downregulating their ability for migration and plasticity, leading to neuro-developmental disorders.

# Appendix

## Stochastic model

In the model, each of the available motors can be found in three different configurations (or motor states): a detached state $D$ and two force-generating attached states $A_1$ (low force-generating state) and $A_2$ (high force-generating state). The possible transitions between distinct allowed motor states are displayed in the following kinetic scheme:

$$D \underset{k_{-1}}{\overset{k_1}{\rightleftharpoons}} A_1 \underset{k_{-2}}{\overset{k_2}{\rightleftharpoons}} A_2 \overset{k_3}{\rightarrow} D$$

The rate constants $k_1$, $k_{-1}$, $k_2$, $k_{-2}$ and $k_3$ represent the probability per unit of time for the single reaction to occur, and are expressed in $s^{-1}$. We define the number of motors in the detached configuration, at time $t$, as $n_D(t)$, where $n_1(t)$ and $n_2(t)$ stand for the number of attached motors in configurations $A_1$ and $A_2$, respectively. The number of motors available for the interaction with the actin filament, $N$, is fixed in accordance with the estimate of the number of rupture events in ATP-free solution, so we have: $N = n_D + n_1 + n_2$, and the state of the system is described, at any time $t$, by the two-dimensional vector $\boldsymbol{n}(t) = (n_1(t), n_2(t))$.

The microscopic dynamics of the system are stochastic due to the probabilistic nature of the motor transitions in the kinetic scheme. The time evolution of the probability distribution $P(n, t)$ of finding the system in the state $\boldsymbol{n}$ at time $t$ is ruled by a master equation.

In the mean field analysis, we focus on the average profile of the ensemble force; that is, disregarding the fluctuations, we defined the fractions of motors in states $A_1$ and $A_2$ as $y$ and $z$. The deterministic evolution of $y$ and $z$ can be obtained from the solution of the master equation. As the experimental data sets were obtained at room temperature, we worked under the adiabatic approximation introduced in Buonfiglio et al. (2024). More specifically, we neglected the contribution of motors in the $A_1$ configuration [see supplementary information in Buonfiglio et al. (2024) for a validation of this approach]. This approximation allowed us to obtain a simple description of the

attachment/force-generation/detachment events in terms of the fraction $z$ of motors in the high-force configuration. For the system under examination, we also defined the duty ratio $r$ as the fraction of attached motors. We noted that, since in the case at hand motors are prevalently found in state $A_2$, a straightforward calculation yields: $z^* = r$, where $z^*$ is the average concentration of motors in state $A_2$ at the isometric force plateau. According to this simplified scheme, we also estimate $\varphi$, the effective rate of ATP consumption, as the flux of motors through the cycle per unit time, i.e. the rate of motors in $A_2$ detaching from actin.

We proceed by estimating the theoretical benchmark for the force exerted by the ensemble of motors working in the stationary state, at the isometric force plateau. To do so we combined the contribution of each individual motor of the collection, labelling the force exerted by motors in state $A_1$ and $A_2$ as $f_1$ and $f_2$, respectively. According to the experimental setup, motors were deposited on the surface with a random orientation, with respect to the actin filament. As a consequence, we assumed that the intensity of the motor force depends on the binding angle with the actin monomer. Labelling as $f_0$ the force of a motor 'correctly' oriented (with the physiological orientation with respect to the actin filament), we assumed that the force exerted by a motor decreases up to a minimum value of $0.1 f_0$ (in accordance with Ishijima et al., 1996). Following Pertici et al. (2020) (see their supplementary fig. 2a) we postulate that the force $f_1$ is a random variable, uniformly distributed within the interval $I_1 = [-f_0, f_0]$, while the force $f_2$ is randomly extracted from the interval $I_2 = [0.1 f_0, f_0]$. At each time, the mean force exerted by the ensemble of fixed size $N$ of myosin motors is:

$$\langle F(t) \rangle = \langle n_1(t) \rangle \langle f_1 \rangle + \langle n_2(t) \rangle \langle f_2 \rangle = \frac{11}{20} f_0 N z^*$$

given that $\langle f_1 \rangle = 0$ and $\langle f_2 \rangle = (11/20) f_0$. In the stationary state, the force of the ensemble converges to the asymptotic plateau $F_0$, and thus:

$$F_0 = \frac{11}{20} f_0 N r$$

where use has been made of the condition $z^* = r$. As readily follows from the above relation, one cannot recover closed estimates for the motor force $f_0$ and the duty ratio $r$ from direct measurements of the average asymptotic force exerted by the ensemble of active motors. Indeed, we can only conclude that $f_0$ and $r$ are constrained to fall on a hyperbole that is univocally determined by the measured value of $F_0$ (see Fig. 4, solid and dashed lines). For this reason, we took into consideration the fluctuations superimposed on $F_0$, and by properly gauging the stochastic component of the dynamics, we were able to resolve the above-mentioned degeneracy [see Buonfiglio et al. (2024) for the details of the characterization of the stochastic dynamics of an ensemble of $N$ molecular motors]. The emerging collective force is indeed subjected to finite size fluctuations. The proposed procedure requires solving the master equation, with emphasis on the stationary probability distribution $P^{st}$, a task that is accomplished in Buonfiglio et al. (2024). By using the recovered expression for $P^{st}$ as a function of the model parameters, one could compute $P(F)$, the distribution of the exerted force $F$, which in turn depends on the kinetic constants and on the maximum force $f_0$ exerted by individual motors. The theoretical expression for $P(F)$ can be compared with the homologous experimental curve, as obtained from the recorded time series of the force of the ensemble. With an optimization procedure based on the stochastic simulated annealing algorithm (Cardoso et al., 1996), information on the underlying parameters of the dynamics was recovered. In particular, we proved that the information stemming from the finite-size fluctuations

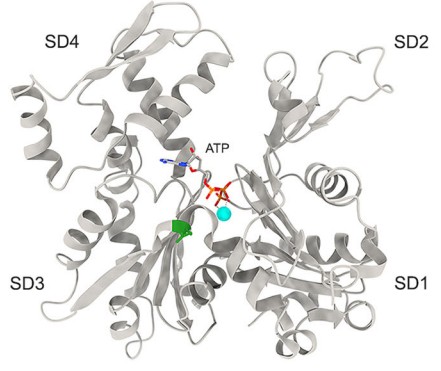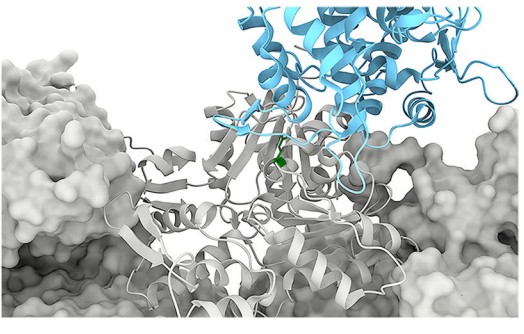

**Figure A1. Homology model of human $\gamma$-actin**
Left, $\gamma$-actin monomer. ATP is bound to the actin nucleotide-binding site and the coordinated $Mg^{2+}$ ion is coloured in cyan. The mutated residue E334 is shown in green. Model is based on PDB-ID 2BTF. Right, the position of the mutated residue E334 in the acto-myosin complex ($\gamma$-actin, grey; NM IIC, light blue) is shown in green (PDB-ID: 5JLH).

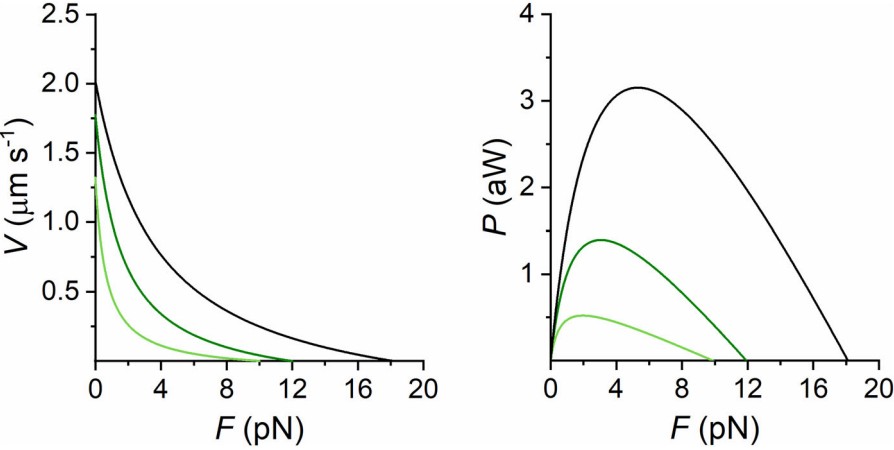

**Figure A2. Comparison of the *F–V* and *P–F* relations**
Superimposition of the *F–V* (left) and *P–F* (right) relations obtained from the nanomachine experiments performed with WT, p.E334Q γ-actin filaments and heterofilaments interacting with the ensemble of psoas HMM. *F–V* and *P–F* relations for p.E334Q γ-actin and heterofilaments are Hill's fits to the experimental data shown in Fig. 2*C* and *D*.

is essential to unambiguously determine both the duty ratio and the single motor force (see supplementary information in Buonfiglio et al., 2024) (Table 3 and Fig. 4). As previously mentioned, we are also able to estimate the effective flux of motors through the cycle per unit time $\varphi$ (Table 3).

Figure A1–A2

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

## Additional information

### Data availability statement

All relevant data, protocols and materials are included in the paper. The data generated and analysed during this study will

be available from the corresponding author upon reasonable request.

## Competing interests

The authors declare that no competing interests exist.

## Author contributions

I.P. and P.B. designed the study; I.P. and P.B. performed the experiments; I.P., E.B., D.J.M. and P.B. analysed and interpreted the data; J.N.G. produced and purified the recombinant actin; V.B. and D.F. implemented the stochastic model simulation; I.P. wrote the paper; all authors revised the manuscript for its conceptual content.

## Funding

This research was supported by the European Joint Programme on Rare Diseases 2019 (PredACTINg, EJPRD19–033), funded by the European Union's Horizon 2020 research and innovation programme under grant agreement N°825575 with support from the Italian Ministry of University and Research (DM 1638) and the German Federal Ministry of Education and Research (01GM1922B), and by the Italian Ministry of University and Research (PRIN 2022, MAS-NeurActin, 2022XJ29R7). J.N.G. is supported by the PREPARE programme for medical scientists from Hannover Medical School.

## Acknowledgements

The authors thank all the members of the PredACTINg consortium for valuable discussions and Michela Moricci for support in setting up the heterofilament experiments. The authors are grateful to Vincenzo Lombardi for relevant comments on the manuscript.

Open access publishing facilitated by Universita degli Studi di Firenze, as part of the Wiley - CRUI-CARE agreement.

## Keywords

actinopathies, cytoskeletal actin, cytoskeletal actin mutants

## Supporting information

Additional supporting information can be found online in the Supporting Information section at the end of the HTML view of the article. Supporting information files available:

**Peer Review History**

