## [Peer Review History · The Journal of Physiology]

Mechanics of blunting of actin-myosin interaction dynamics by the actinopathy-causing mutation E334Q in cytoskeletal γ -actin

Irene Pertici, Valentina Buonfiglio, Johannes N. Greve, Elena Battirossi, Duccio Fanelli, Dietmar J. Manstein, and Pasquale Bianco

DOI: 10.1113/JP289622

Corresponding author(s): Irene Pertici (irene.pertici@unifi.it)

Review Timeline:

Submission Date:	07-Jul-2025
Editorial Decision:	08-Aug-2025
Revision Received:	26-Sep-2025
Editorial Decision:	20-Oct-2025
Revision Received:	23-Oct-2025
Editorial Decision:	24-Oct-2025
Revision Received:	24-Oct-2025
Accepted:	27-Oct-2025

Senior Editor: Paul Greenhaff

Reviewing Editor: Wolfgang Linke

Transaction Report:

Dear Dr Pertici,

Re: JP-RP-2025-289622 "**Mechanics of blunting of actin-myosin interaction dynamics by the actinopathy-causing mutation E334Q in cytoskeletal γ -actin**" by Irene Pertici, Valentina Buonfiglio, Johannes N. Greve, Elena Battirosi, Duccio Fanelli, Dietmar J. Manstein, and Pasquale Bianco

Thank you for submitting your manuscript to The Journal of Physiology. It has been assessed by a Reviewing Editor and by 2 expert referees and we are pleased to tell you that it is potentially acceptable for publication following satisfactory major revision.

REVISION CHECKLIST:

We look forward to receiving your revised submission.

Yours sincerely,

Paul Greenhaff
Senior Editor
The Journal of Physiology

REQUIRED ITEMS

- 1) - Author photo and profile. First or joint first authors are asked to provide a short biography (no more than 100 words for one author or 150 words in total for joint first authors) and a portrait photograph. These should be uploaded and clearly labelled together in a Word document with the revised version of the manuscript. See Information for Authors for further details.
- 2) - You must start the Methods section with a paragraph headed Ethical approval (https://jp.msubmit.net/cgi-bin/main.plex?form_type=display_requirements#methods).
 - Research must comply with The Journal's policies regarding animal experiments (<https://physoc.onlinelibrary.wiley.com/hub/animal-experiments>) and adherence to these policies must be stated in the manuscript.
 - Authors should confirm in their Methods section that their experiments were carried out according to the guidelines laid down by their institution's animal welfare committee, including an ethics approval reference number. The Methods section must contain a statement about access to food, water and housing, details of the anaesthetic regime: anaesthetic used, dose and route of administration, and method of killing the experimental animals.
- 3) - Please upload separate high-quality figure files via the submission form.
- 4) - Papers must comply with the Statistics Policy: https://jp.msubmit.net/cgi-bin/main.plex?form_type=display_requirements#statistics.

In summary:

- If $n \leq 30$, all data points must be plotted in the figure in a way that reveals their range and distribution. A bar graph with data points overlaid, a box and whisker plot or a violin plot (preferably with data points included) are acceptable formats.
- If $n > 30$, then the entire raw dataset must be made available either as supporting information, or hosted on a not-for-profit repository, e.g. FigShare, with access details provided in the manuscript.
- 'n' clearly defined (e.g. x cells from y slices in z animals) in the Methods. Authors should be mindful of pseudoreplication.
- All relevant 'n' values must be clearly stated in the main text, figures and tables.
- The most appropriate summary statistic (e.g. mean or median and standard deviation) must be used. Standard Error of the Mean (SEM) is not permitted.
- Exact p values must be stated. Authors must not use 'greater than' or 'less than'. Exact p values must be stated to three significant figures even when 'no statistical significance' is claimed.

5) - Please include an Abstract Figure file, as well as the Figure Legend text within the main article file. The Abstract Figure is a piece of artwork designed to give readers an immediate understanding of the research and should summarise the main conclusions. If possible, the image should be easily 'readable' from left to right or top to bottom. It should show the physiological relevance of the manuscript so readers can assess the importance and content of its findings. Abstract Figures should not merely recapitulate other figures in the manuscript. Please try to keep the diagram as simple as possible and without superfluous information that may distract from the main conclusion(s). Abstract Figures must be provided by authors no later than the revised manuscript stage and should be uploaded as a separate file during online submission labelled as File Type 'Abstract Figure'. Please also ensure that you include the figure legend in the main article file. All Abstract Figures should be created using BioRender. Authors should use The Journal's premium BioRender account to export high-resolution images. Details on how to use and access the premium account are included as part of this email.

EDITOR COMMENTS

Reviewing Editor:

The two experts commented on your paper with varying levels of enthusiasm. Both recognized the value of your study but also raised important points that should be addressed in detail before we can further consider your manuscript for publication in The Journal. In addition, please provide the missing information regarding animal ethics.

Senior Editor:

Thank you for the manuscript submission to The Journal of Physiology which has been considered by a reviewing editor and two expert reviewers. As can be seen from the feedback provided there is a divergent opinion about the merits of the work, with significant concerns from Reviewer 1 about the statistic employed and the layout and structure of the paper (please consult the Instructions for Authors on The Journal of Physiology web site).

The Reviewing Editor believes the manuscript could be impactful but major revisions are required for it to be considered further. Of additional notable importance, The Journal of Physiology has clear guidelines for the undertaking and reporting of statistical analysis and animal ethics, the manuscript does not appear to currently adhere to these guidelines. Please consult both sets of guidelines when revising the manuscript. We look forward to receiving the revised manuscript in due course. Please address all comments provided on a point-by-point basis. Thank you.

The statistics policy, animal procedures and the overall structure of the paper need to comply with The Journal requirements.

REFEREE COMMENTS

Referee #1:

Pertici et al. in 2018 have devised a nanomachine device to study myosin actin interaction. Myosin dimers are purified from rabbit psoas muscles. The authors refined the device in 2020 to include investigation of Ca²⁺-dependent regulatory processes. In 2021, Pertici et al. used the device to study hind-limb muscles from the frog. In 2024, the authors compared fast (psoas) and slow (soleus) twitch muscles of the rabbit. In this manuscript in 2025, Pertici et al. used their nanomachine to study actin filaments comparing wildtype and mutant monomers.

1. **Abstract:** A sentence or two on the methods is missing. Currently, the method sentence is vaguely described as "... we employ a nanomachine ...". Describe, in a sentence or two, in the Abstract, the methods used in producing data in Figures 1-4.

2. **Key Point Summary:** A key point on the general/overall findings is missing. As written, it is only stated as "we ... demonstrated how they are affected by ... mutation". How were they affected, in lay terms?

3. **Introduction:** References are missing in the sentence stating that "So far, the effects of E334Q mutation on the interaction dynamics of cytoskeletal γ -actin have been studied only under unloaded conditions." Also, elaborate on this statement, i.e, what would be some limitations studying under unloaded conditions; what are the methodologies used in current studies of E334Q mutation.

4. **"In situ" loads:** The Abstract stated that the experiments were conducted that mimics "in situ" loading conditions. What are the loading conditions that the authors mimic?

5. **Methods:** HMM is never defined in this manuscript.

6. **Methods:** The Methods section largely describes the nanomachine itself and the stochastic model (both of which have been described by the authors in their previous papers). The Methods section is missing the experimental protocols/interventions that are specific to the present study that obtained the data plotted in Figures 1-4.
7. **Results:** The Results section blends methods and results. Perhaps the authors find it more appropriate to describe methods in the Results section? If so, the Methods section suffers from lack of details of the experimental protocols.
8. **Figure 1:** In Panels B and C, show the data at [HMM] = 0.3 mg/mL.
9. **Figure 1:** In Panels B-D, the y-axis labelled 'd' is not defined. What is 'd'? Can 'd' be illustrated in Panel A?
10. **Figure 1:** Both the x-axis and y-axis should be consistent for ease of comparison across all three [HMM] cases. Currently, for example, in Figure 1C, the time-axis ends at about 25 s for the 0.1 mg/mL case but at 40 s for the 0.2 mg/mL case? The same for the y-axis.
11. **Figure 1:** In both concentrations of HMM, shouldn't 'd' (on the y-axis) be the same, or otherwise the 'd' was ended shorter in one case than the other, leaving the reader to wonder whether there could be additional rupture events later in time? How do the readers know whether the actin filament has detached completely, given the different time axis between panels? The same and consistent y-axis range is important because rupture event can still occur following a long period of zero force (as shown in Panel D).
12. **Figure 2:** In Panel A, the phases 4-6 are not shown/drawn?
13. **Figure 2:** Consistent axes across all cases are needed.
14. **Figure 2:** In Panel C, why did you bin the V-F data? Also, was Hill's hyperbolic fit applied to the individual data points or to the bin average?
15. **Figure 2:** Add a new Panel D to show individual data points of the P-F relation and fit. Currently, the P-F plots are embedded as insets in Panel C without any data points.
16. **Figure 5:** This figure is redundant. It can be removed when the axes on Figure 2 are consistent across all cases.
17. **Rupture event:** What is the justification for defining a rupture event as "force drops > 2 pN, complete in < 50 ms and not rapidly reversed"? In Figure 1, several rupture events have the force dropped to zero, while some dropped only by a few pN. Also, some force drops appear to be rapidly reversible, i.e., Figure 1C the 1st event, the 3rd and the 4th.
18. **Discussion:** As written, the Discussion section is missing key aspects of what constitute a Discussion. As written, the 1st paragraph is a summary of the study, which is nicely written and is appropriate. The 2nd paragraph, however, is a repeat of the Results section. Interestingly, it only cites one literature paper (Greve et al., 2024). The authors should consider expanding the Discussion to discuss, for example, (i) the link between their measurement of shortening velocity to those obtained under unloaded conditions (as introduced in the Introduction); (ii) the application of their technique and data to interpret cellular processes associated with cytoskeletal actin filaments (i.e., cytokinesis, apoptosis and cell locomotion, as stated in the first sentence of the Abstract); (iii) the perspective of applying their measurements to understand those pathology conditions (as stated in Key Point #1); (iv) how do their measurements inform attachment detachment rate constants and kinetics (currently, it is stated in line 415, "the reduction of the kinetic parameters", but by how much?); and (v) the limitation of their study.
19. **Statistics:** With limited information in the Statistical Analysis section, it is unclear what statistical test was used in all Tables 1-3. It appears that "Student's t test was usually employed for statistical testing" (as stated in the Statistical Analysis section). If t-test was used, then it is inappropriate. Specifically, in Table 1, 2-way ANOVA would be more appropriate: factor 'HMM concentration' (0.1, 0.2 and 0.3 mg/mL) and factor 'group' (WT, p.E334Q, and Heterofilaments). In Tables 2-3, ANOVA would be more appropriate. Note also that for Journal of Physiology, the policy for statistics includes clearly stating all 'n' values and p-values in the figures and their legends.

Referee #2:

Pertici et al. report on in vitro functional assays with actin filaments containing WT γ -actin, a variant (E334Q) that is associated with inherited, non-muscle actinopathies, or a 50:50 mixture of WT:E334Q variant. In the variant, a polar but uncharged sidechain (Q) replaces a negatively charged sidechain (E) that is important for ionic interaction with positively charged residues in the "cardiomyopathy loop" of myosin. The authors used in vitro motility assay (unloaded filament sliding) and their central approach, the Florence "nanomachine" assay (loaded). The authors used fast skeletal muscle myosin rather than the biologically relevant myosin (non-muscle myosin II), but this was justified for technical reasons. All functional parameters were lower with 50% E334Q and were lower still with 100% E334Q, with almost all of the differences being statistically significant. The authors conclude that the observed defects in brain and neural development are very likely due, at least in part, to the impaired function when non-muscle myosin interacts with filaments containing mutant γ -actin in neuronal cells.

Major:

This is an interesting study conducted by a team of experts that yielded results with implications for the role of actomyosin in development and mechanisms for actinopathies.

Regarding the 50:50 mixture of WT and variant actin to generate hetero-filaments (e.g., lines 124 and 274-5): This is an important experiment, but how certain are the authors that the patient's cells contain a 50:50 mix of WT and variant actin protein? As the authors are aware, protein expression levels in cardiomyopathy patients often do not correspond to the expected 50:50 mixture that would be predicted simply from genetics. The cited reference (Di Donato et al., 2016) appears to be quite clinical in its focus.

Regarding the actin filaments, the authors did not include tropomyosin. At minimum, this should be addressed in a "Limitations" section of discussion.

Minor:

Line 169: "...consisted in an..." -> "...consisted of an..."

Lines 223-228: specify what statistical software was used. R?

Motility assays and results: There is no description of "conventional" in vitro motility assay methods in the Methods section. It was also challenging to find any description of the motility assay results that are included in Table 2 (Vf). The brief, relevant text is in the middle of a paragraph in Results (line 310).

END OF COMMENTS

Pertici et al. in 2018 have devised a nanomachine device to study myosin-actin interaction. Myosin dimers are purified from rabbit psoas muscles. The authors refined the device in 2020 to include investigation of Ca^{2+} -dependent regulatory processes. In 2021, Pertici et al. used the device to study hind-limb muscles from the frog. In 2024, the authors compared fast (psoas) and slow (soleus) twitch muscles of the rabbit. In this manuscript in 2025, Pertici et al. used their nanomachine to study actin filaments comparing wildtype and mutant monomers.

1. **Abstract:** A sentence or two on the methods is missing. Currently, the method sentence is vaguely described as "... we employ a nanomachine ...". Describe, in a sentence or two, in the Abstract, the methods used in producing data in Figures 1-4.
2. **Key Point Summary:** A key point on the general/overall findings is missing. As written, it is only stated as "we ... demonstrated how they are affected by ... mutation". How were they affected, in lay terms?
3. **Introduction:** References are missing in the sentence stating that "So far, the effects of E334Q mutation on the interaction dynamics of cytoskeletal γ -actin have been studied only under unloaded conditions." Also, elaborate on this statement, i.e, what would be some limitations studying under unloaded conditions; what are the methodologies used in current studies of E334Q mutation.
4. **"In situ" loads:** The Abstract stated that the experiments were conducted that mimics "in situ" loading conditions. What are the loading conditions that the authors mimic?
5. **Methods:** HMM is never defined in this manuscript.
6. **Methods:** The Methods section largely describes the nanomachine itself and the stochastic model (both of which have been described by the authors in their previous papers). The Methods section is missing the experimental protocols/interventions that are specific to the present study that obtained the data plotted in Figures 1-4.
7. **Results:** The Results section blends methods and results. Perhaps the authors find it more appropriate to describe methods in the Results section? If so, the Methods section suffers from lack of details of the experimental protocols.
8. **Figure 1:** In Panels B and C, show the data at $[\text{HMM}] = 0.3 \text{ mg/mL}$.
9. **Figure 1:** In Panels B-D, the y-axis labelled 'd' is not defined. What is 'd'? Can 'd' be illustrated in Panel A?
10. **Figure 1:** Both the x-axis and y-axis should be consistent for ease of comparison across all three $[\text{HMM}]$ cases. Currently, for example, in Figure 1C, the time-axis ends at about 25 s for the 0.1 mg/mL case but at 40 s for the 0.2 mg/mL case? The same for the y-axis.
11. **Figure 1:** In both concentrations of HMM, shouldn't 'd' (on the y-axis) be the same, or otherwise the 'd' was ended shorter in one case than the other, leaving the reader to wonder whether there could be additional rupture events later in time? How do the readers know whether the actin filament has detached completely, given the different time-axis between

panels? The same and consistent y-axis range is important because rupture event can still occur following a long period of zero force (as shown in Panel D).

12. **Figure 2:** In Panel A, the phases 4-6 are not shown/drawn?
13. **Figure 2:** Consistent axes across all cases are needed.
14. **Figure 2:** In Panel C, why did you bin the V-F data? Also, was Hill's hyperbolic fit applied to the individual data points or to the bin average?
15. **Figure 2:** Add a new Panel D to show individual data points of the P-F relation and fit. Currently, the P-F plots are embedded as insets in Panel C without any data points.
16. **Figure 5:** This figure is redundant. It can be removed when the axes on Figure 2 are consistent across all cases.
17. **Rupture event:** What is the justification for defining a rupture event as "force drops > 2 pN, complete in < 50 ms and not rapidly reversed"? In Figure 1, several rupture events have the force dropped to zero, while some dropped only by a few pN. Also, some force drops appear to be rapidly reversible, i.e., Figure 1C the 1st event, the 3rd and the 4th.
18. **Discussion:** As written, the Discussion section is missing key aspects of what constitute a Discussion. As written, the 1st paragraph is a summary of the study, which is nicely written and is appropriate. The 2nd paragraph, however, is a repeat of the Results section. Interestingly, it only cites one literature paper (Greve et al., 2024). The authors should consider expanding the Discussion to discuss, for example, (i) the link between their measurement of shortening velocity to those obtained under unloaded conditions (as introduced in the Introduction); (ii) the application of their technique and data to interpret cellular processes associated with cytoskeletal actin filaments (i.e., cytokinesis, apoptosis and cell locomotion, as stated in the first sentence of the Abstract); (iii) the perspective of applying their measurements to understand those pathology conditions (as stated in Key Point #1); (iv) how do their measurements inform attachment-detachment rate constants and kinetics (currently, it is stated in line 415, "the reduction of the kinetic parameters", but by how much?); and (v) the limitation of their study.
19. **Statistics:** With limited information in the Statistical Analysis section, it is unclear what statistical test was used in all Tables 1-3. It appears that "Student's t test was usually employed for statistical testing" (as stated in the Statistical Analysis section). If t-test was used, then it is inappropriate. Specifically, in Table 1, 2-way ANOVA would be more appropriate: factor 'HMM concentration' (0.1, 0.2 and 0.3 mg/mL) and factor 'group' (WT, p.E334Q, and Heterofilaments). In Tables 2-3, ANOVA would be more appropriate. Note also that for Journal of Physiology, the policy for statistics includes clearly stating all 'n' values and p-values in the figures and their legends.

We thank the Editors and the Referees for the careful review of the manuscript.

We revised the paper according to the comments and points raised by the Referees and to the guidelines of the Journal of Physiology for statistical analysis and animal ethics. All changes are highlighted in yellow in the manuscript text.

As requested by the Journal, we uploaded the Graphical Abstract and incorporated its legend into the main text. Since the model of the γ -actin monomer (from Figure A1) was slightly modified for the purpose of clarity to be shown in the Graphical Abstract, we incorporated these changes into Figure A1, where we added also the modelling of the acto-myosin complex.

Here follows (in red italics) the point-by-point response to Editors and Referees' comments. The line numbers refer to the version of the manuscript with highlighted changes.

EDITOR COMMENTS

Reviewing Editor:

The two experts commented on your paper with varying levels of enthusiasm. Both recognized the value of your study but also raised important points that should be addressed in detail before we can further consider your manuscript for publication in The Journal. In addition, please provide the missing information regarding animal ethics.

Senior Editor:

Thank you for the manuscript submission to The Journal of Physiology which has been considered by a reviewing editor and two expert reviewers. As can be seen from the feedback provided there is a divergent opinion about the merits of the work, with significant concerns from Reviewer 1 about the statistic employed and the layout and structure of the paper (please consult the Instructions for Authors on The Journal of Physiology web site).

The Reviewing Editor believes the manuscript could be impactful but major revisions are required for it to be considered further. Of additional notable importance, The Journal of Physiology has clear guidelines for the undertaking and reporting of statistical analysis and animal ethics, the manuscript does not appear to currently adhere to these guidelines. Please consult both sets of guidelines when revising the manuscript. We look forward to receiving the revised manuscript in due course. Please address all comments provided on a point-by-point basis. Thank you.

The statistics policy, animal procedures and the overall structure of the paper need to comply with The Journal requirements.

We revised the paper according to the Journal of Physiology requirements in terms of statistics policy, animal procedures and manuscript structure.

REFEREE COMMENTS

Referee #1:

Pertici et al. in 2018 have devised a nanomachine device to study myosin actin interaction. Myosin dimers are purified from rabbit psoas muscles. The authors refined the device in 2020 to include investigation of Ca²⁺-dependent regulatory processes. In 2021, Pertici et al. used the device to study hind-limb muscles from the frog. In 2024, the authors compared fast (psoas) and slow (soleus) twitch muscles of the rabbit. In this manuscript in 2025, Pertici et al. used their nanomachine to study actin filaments comparing wildtype and mutant monomers.

1. Abstract: A sentence or two on the methods is missing. Currently, the method sentence is vaguely described as "... we employ a nanomachine ...". Describe, in a sentence or two, in the Abstract, the methods used in producing data in Figures 1-4.

The abstract has been modified with the addition of a synthetic description of the nanomachine and a more explicit mention of the results.

2. Key Point Summary: A key point on the general/overall findings is missing. As written, it is only stated as "we ... demonstrated how they are affected by ... mutation". How were they affected, in lay terms?

The Key Point Summary has been modified accordingly.

3. Introduction: References are missing in the sentence stating that "So far, the effects of E334Q mutation on the interaction dynamics of cytoskeletal γ -actin have been studied only under unloaded conditions." Also, elaborate on this statement, i.e, what would be some limitations studying under unloaded conditions; what are the methodologies used in current studies of E334Q mutation.

The reference (Greve et al. 2024 Elife, 12, RP93013) has been added at the end of the sentence and the corresponding methodologies mentioned explicitly.

4. "In situ" loads: The Abstract stated that the experiments were conducted that mimics "in situ" loading conditions. What are the loading conditions that the authors mimic?

Our approach allows the mechanical performance of the actin-myosin interaction to be estimated in the whole range of loading conditions that the system may experience in situ, from the absence of load to several loads up to that corresponding to the maximum isometric force. The argument has been revised for clarification.

5. Methods: HMM is never defined in this manuscript.

Definition introduced at line 190.

6. Methods: The Methods section largely describes the nanomachine itself and the stochastic model (both of which have been described by the authors in their previous papers). The Methods section is missing the experimental protocols/interventions that are specific to the present study that obtained the data plotted in Figures 1-4.

7. Results: The Results section blends methods and results. Perhaps the authors find it more appropriate to describe methods in the Results section? If so, the Methods section suffers from lack of details of the experimental protocols.

We reply to questions 6-7 at once since they concern the same topic.

We find it convenient for the reader to have an explanation of the experimental protocol when the corresponding measurements are described, i.e. in the Results section. However, we implemented the Methods section with a brief description of the protocols, and we removed from Results and moved into Methods methodological details (for instance, methods for counting the rupture events, now at lines 263-265 and 274-275; protocol for estimating the transition to the steady isometric force, now at lines 296-298).

Please note however that no significant changes to the original protocols (Pertici et al. 2018 Nat Commun 9:3532; Pertici et al. 2020 Int J Mol Sci 21:7372) were made for the present study.

8. Figure 1: In Panels B and C, show the data at [HMM] = 0.3 mg/mL.

Data at [HMM] = 0.3 mg/mL have been added also for panel C and panel D, and the statistics reported in Table 1.

9. Figure 1: In Panels B-D, the y-axis labelled 'd' is not defined. What is 'd'? Can 'd' be illustrated in Panel A?

10. Figure 1: Both the x-axis and y-axis should be consistent for ease of comparison across all three [HMM] cases. Currently, for example, in Figure 1C, the time-axis ends at about 25 s for the 0.1 mg/mL case but at 40 s for the 0.2 mg/mL case? The same for the y-axis.

11. Figure 1: In both concentrations of HMM, shouldn't 'd' (on the y-axis) be the same, or otherwise the 'd' was ended shorter in one case than the other, leaving the reader to wonder whether there could be additional rupture events later in time? How do the readers know whether the actin filament has detached completely, given the different time axis between panels? The same and consistent y-axis range is important because rupture event can still occur following a long period of zero force (as shown in Panel D).

We answer the questions 9-11 together since they all concern the same Figure.

In the previous version of Figure 1, the traces were cut a few seconds after the last rupture for a better resolution in time of the rupture events. Following the Referee's concerns, we show now all the traces in panels B-D up to 80 s (> 30 s from the last rupture event). To keep constant the x-axis implies a reduction in time resolution of the events, but our analysis is done on expanded traces.

In the old Figure 1, the movement imposed to the nanopositioner ("d") was shown in the upper panel of all traces. In the new version, all the traces end at 80 s (corresponding to a total movement of 4 μm), so the nanopositioner movement signal would be redundant. For this, Figure 1 shows now only F_x over time at the various concentrations and conditions indicated.

The amplitudes of the force drops (as shown by the y-axis) depend on the distance between consecutive myosin motors, which are randomly distributed on the pipette lateral surface (lines 274-275). For this reason, we preferred to maintain the optimal y-range for rupture visualization in each graph.

12. Figure 2: In Panel A, the phases 4-6 are not shown/drawn?

We have clarified the issue in the figure legend by identifying the phases depicted in A with the corresponding phases in B.

13. Figure 2: Consistent axes across all cases are needed.

Axes are now consistent.

14. Figure 2: In Panel C, why did you bin the V-F data? Also, was Hill's hyperbolic fit applied to the individual data points or to the bin average?

We thank the Referee for identifying the missing information, which has been integrated into the Methods section (lines 284-292).

15. Figure 2: Add a new Panel D to show individual data points of the P-F relation and fit. Currently, the P-F plots are embedded as insets in Panel C without any data points.

P-F relations are now shown in a dedicated panel of Fig. 2 (panel D). Note that the P-F relation is obtained directly from the fit of the corresponding F-V relation (panel C), by multiplying by F the V value calculated by the fit. For this reason, we find it more appropriate to show only the calculated P-F relations.

16. Figure 5: This figure is redundant. It can be removed when the axes on Figure 2 are consistent across all cases.

The Figure has been moved to the Appendix as Figure A2, as we believe that it provides the most immediate and simple visual understanding of the effect of the mutation on the F-V parameters.

17. Rupture event: What is the justification for defining a rupture event as "force drops > 2 pN, complete in < 50 ms and not rapidly reversed"? In Figure 1, several rupture events have the force dropped to zero, while some dropped only by a few pN. Also, some force drops appear to be rapidly reversible, i.e., Figure 1C the 1st event, the 3rd and the 4th.

The definition of rupture event was employed since the first paper featuring the synthetic nanomachine (Pertici et al. 2018 Nat Commun 9:3532) and takes into account the instrumental constraints. In Methods (lines 267-273) we provided details of the selection criteria.

18. Discussion: As written, the Discussion section is missing key aspects of what constitute a Discussion. As written, the 1st paragraph is a summary of the study, which is nicely written and is appropriate. The 2nd paragraph, however, is a repeat of the Results section. Interestingly, it only cites one literature paper (Greve et al., 2024). The authors should consider expanding the Discussion to discuss, for example, (i) the link between their measurement of shortening velocity to those obtained under unloaded conditions (as introduced in the Introduction); (ii) the application of their technique and data to interpret cellular processes associated with cytoskeletal actin filaments (i.e., cytokinesis, apoptosis and cell locomotion, as stated in the first sentence of the Abstract); (iii) the perspective of applying their measurements to understand those pathology conditions (as stated in Key Point #1); (iv) how do their measurements inform attachment detachment rate constants and kinetics (currently, it is stated in line 415, "the reduction of the kinetic parameters", but by how much?); and (v) the limitation of their study.

The paucity of cited literature in the Discussion section reflects the lack of functional characterization of the effects of Non-Muscle Actinopathy-associated mutations as p.E334Q at the molecular level. As of today, the only non-clinical study regarding this particular mutation is its biochemical characterization published by two of the coauthors of this paper (Greve et al. 2024 Elife, 12, RP93013). However, we implemented the Discussion section with the comparison between V_f and V_o (lines 521-524), the quantitative effect of the mutation on the kinetic parameters (line 534), the putative effects of the loss of basic function described here on cellular processes based on cell motility (lines 538-543) and study limitations (absence of tropomyosin, lines 554-561; heterofilament model, lines 546-553).

19. Statistics: With limited information in the Statistical Analysis section, it is unclear what statistical test was used in all Tables 1-3. It appears that "Student's t test was usually employed for statistical testing" (as stated in the Statistical Analysis section). If t-test was used, then it is inappropriate. Specifically, in Table 1, 2-way ANOVA would be more appropriate: factor 'HMM concentration' (0.1, 0.2 and 0.3 mg/mL) and factor 'group' (WT, p.E334Q, and Heterofilaments). In Tables 2-3, ANOVA would be more appropriate. Note also that for Journal of Physiology, the policy for statistics includes clearly stating all 'n' values and p-values in the figures and their legends.

We thank the Referee for his/her suggestions regarding Statistics. We implemented the Statistical Analysis section, table and legends as requested by the Journal guidelines and performed one-way or two-way ANOVA for the statistical analysis of the data shown in Tables 1-3.

Referee #2:

Pertici et al. report on in vitro functional assays with actin filaments containing WT γ -actin, a variant (E334Q) that is associated with inherited, non-muscle actinopathies, or a 50:50 mixture of WT:E334Q variant. In the variant, a polar but uncharged sidechain (Q) replaces a negatively charged sidechain (E) that is important for ionic interaction with positively charged residues in the "cardiomyopathy loop" of myosin. The authors used in vitro motility assay (unloaded filament sliding) and their central approach, the Florence "nanomachine" assay (loaded). The authors used fast skeletal muscle myosin rather than the biologically relevant myosin (non-muscle myosin II), but this was justified for technical reasons. All functional parameters were lower with 50% E334Q and were lower still with 100% E334Q, with almost all of the differences being statistically significant. The authors conclude that the observed defects in brain and neural development are very likely due, at least in part, to the impaired function when non-muscle myosin interacts with filaments containing mutant γ -actin in neuronal cells.

Major:

This is an interesting study conducted by a team of experts that yielded results with implications for the role of actomyosin in development and mechanisms for actinopathies.

We thank the Referee for his/her understanding of the achievement by the nanomachine-based approach as far as the mechanical characterization of actin-myosin interaction with WT and mutant γ -actin.

Regarding the 50:50 mixture of WT and variant actin to generate hetero-filaments (e.g., lines 124 and 274-5): This is an important experiment, but how certain are the authors that the patient's cells contain a 50:50 mix of WT and variant actin protein? As the authors are aware, protein expression levels in cardiomyopathy patients often do not correspond to the expected 50:50 mixture that would be predicted simply from genetics. The cited reference (Di Donato et al., 2016) appears to be quite clinical in its focus.

The referee is right in pointing out the limits of our heterofilament construct in resembling the actin filament conditions in patients' cells, since some uncharacterized mechanism regulating mutant protein levels in the cell may occur. However, we think that the 1:1 condition can be considered as the most informative approximation: previous studies showed that the p.E334Q mutation does not affect the folding and the stability of the monomer and of the filamentous actin (Greve et al. 2024 Elife, 12, RP93013), thus likely excluding the mutant actin degradation as the proposed disease mechanism, and favouring the hypothesis that the pathological outcome arises from p.E334Q actin integration into the

cytoskeleton, where it shows defects in the interaction with multiple binding partners, including myosin. We included an explicit mention to this in the Discussion section (lines 546-553).

Regarding the actin filaments, the authors did not include tropomyosin. At minimum, this should be addressed in a "Limitations" section of discussion.

The Discussion now comprises the consideration of the limits of tropomyosin absence and the perspective to introduce it in the system under study (lines 554-561).

Minor:

Line 169: "...consisted in an..." -> "...consisted of an..."

Amended.

Lines 223-228: specify what statistical software was used. R?

The name of the software used for statistical analysis is now specified in the "Statistical analysis" section.

Motility assays and results: There is no description of "conventional" in vitro motility assay methods in the Methods section. It was also challenging to find any description of the motility assay results that are included in Table 2 (V_f). The brief, relevant text is in the middle of a paragraph in Results (line 310).

We added a description of conventional in vitro motility assay (IVMA) in a separate paragraph in the Methods section. Since in this study IVMA was performed only for initial protein quality check, we do not think its results are worth a dedicated paragraph in the Results section. However, the motility assay results are found in the Results paragraph starting at line 399, as we find convenient to compare the filament sliding velocity measured with IVMA (V_f), commonly referred to as "unloaded", to the truly unloaded maximum sliding velocity (V_0) obtained from the F-V relation. In the revised version of the manuscript, we discussed the reason for the discrepancy between V_f and V_0 (lines 521-524).

Dear Dr Pertici,

Re: JP-RP-2025-289622R1 "**Mechanics of blunting of actin-myosin interaction dynamics by the actinopathy-causing mutation E334Q in cytoskeletal γ -actin**" by Irene Pertici, Valentina Buonfiglio, Johannes N. Greve, Elena Battirosi, Duccio Fanelli, Dietmar J. Manstein, and Pasquale Bianco

Thank you for submitting your manuscript to The Journal of Physiology. It has been assessed by a Reviewing Editor and by 2 expert referees and we are pleased to tell you that it is acceptable for publication following satisfactory revision.

REVISION CHECKLIST:

Please upload two versions of your manuscript text: one with all relevant changes highlighted and one clean version with no changes tracked. The manuscript file should include all tables and figure legends, but each figure/graph should be uploaded as separate, high-resolution files. The journal is now integrated with Wiley's Image Checking service. For further details, see: <https://www.wiley.com/en-us/network/publishing/research-publishing/trending-stories/upholding-image-integrity-wileys->

image-screening-service

We look forward to receiving your revised submission.

Yours sincerely,

Paul Greenhaff
Senior Editor
The Journal of Physiology

EDITOR COMMENTS

Reviewing Editor:

Thank you for providing a responsive revision and answering the statistics points. Both reviewers now agree on the value of your manuscript.

Senior Editor:

Thank you for revising the manuscript in line with the reviewer comments. I am happy to convey the message that the Reviewing Editor and reviewers are of the opinion that the authors have done a good job when revising the manuscript such that it is deemed acceptable for publication. One small point, where possible please illustrate significant differences in the figures which will make the findings easier to interpret for the reader. Thank you.

REFEREE COMMENTS

Referee #1:

The authors have addressed my comments satisfactorily. The layout and structure of the manuscript are good, the Discussion is expanded, and the figures are now complete. The statistical analyses are now appropriate - however, in addition to reporting p-values, I believe that including statistical significance symbols (*) are still required, in all figures and tables (I defer this decision to the journal editors).

Referee #2:

In this revision, the authors have fully addressed my comments on the original submission.

END OF COMMENTS

We thank the Editors and the Referees for their favorable opinion on the revised version of the manuscript. Our comments are provided in red below.

EDITOR COMMENTS

Reviewing Editor:

Thank you for providing a responsive revision and answering the statistics points. Both reviewers now agree on the value of your manuscript.

Senior Editor:

Thank you for revising the manuscript in line with the reviewer comments. I am happy to convey the message that the Reviewing Editor and reviewers are of the opinion that the authors have done a good job when revising the manuscript such that it is deemed acceptable for publication. One small point, where possible please illustrate significant differences in the figures which will make the findings easier to interpret for the reader. Thank you.

We do not find it possible to show significant differences directly in the figures, as Figures 1, 2 and 3 show experimental traces and Figures 2 and 4 report the plots from which we extract the parameters provided in the respective tables with their significance analysis.

REFEREE COMMENTS

Referee #1:

The authors have addressed my comments satisfactorily. The layout and structure of the manuscript are good, the Discussion is expanded, and the figures are now complete. The statistical analyses are now appropriate - however, in addition to reporting p-values, I believe that including statistical significance symbols (*) are still required, in all figures and tables (I defer this decision to the journal editors).

*In the Tables, besides the indication of the p-value, we previously showed the differences in bold when statistically significant ($p < 0.05$). Following the concerns raised by the Referee and the Editor, we have indicated those significant differences as follows: *, $p < 0.05$; **, $p < 0.01$; ***, $p < 0.001$. The changes are highlighted in yellow in the manuscript file.*

Referee #2:

In this revision, the authors have fully addressed my comments on the original submission.

Dear Dr Pertici,

Re: JP-RP-2025-289622R2 "**Mechanics of blunting of actin-myosin interaction dynamics by the actinopathy-causing mutation E334Q in cytoskeletal γ -actin**" by Irene Pertici, Valentina Buonfiglio, Johannes N. Greve, Elena Battirosi, Duccio Fanelli, Dietmar J. Manstein, and Pasquale Bianco

Thank you for submitting your manuscript to The Journal of Physiology. It has been assessed by a Senior Editor and we are pleased to tell you that it is acceptable for publication following satisfactory revision.

REVISION CHECKLIST:

Please upload two versions of your manuscript text: one with all relevant changes highlighted and one clean version with no changes tracked. The manuscript file should include all tables and figure legends, but each figure/graph should be uploaded as separate, high-resolution files. The journal is now integrated with Wiley's Image Checking service. For further details, see: <https://www.wiley.com/en-us/network/publishing/research-publishing/trending-stories/upholding-image-integrity-wileys->

image-screening-service

We look forward to receiving your revised submission.

Yours sincerely,

Paul Greenhaff
Senior Editor
The Journal of Physiology

EDITOR COMMENTS

Senior Editor:

Statistics:

'Regarding the request to illustrate significant differences in the figures and tables, please note that in the tables we already identified as a clear choice for the reader to show these differences in bold when statistically significant ($p < 0.05$). We do not find it possible to show significant differences directly in the figures, as Figs. 1, 2 and 3 show experimental traces and Figs. 2 and 4 report the plots whose extracted parameters are provided in the respective tables with their significance analysis.'

Thank you for clarifying. It would be helpful if the authors could include a statement in the legend to Figs. 2 and 4 and/or the tables to clarify that statistical outcomes pertaining to Figs. 2 and 4 are provided in the tables.

END OF COMMENTS

EDITOR COMMENTS

Senior Editor:

Statistics:

'Regarding the request to illustrate significant differences in the figures and tables, please note that in the tables we already identified as a clear choice for the reader to show these differences in bold when statistically significant ($p < 0.05$). We do not find it possible to show significant differences directly in the figures, as Figs. 1, 2 and 3 show experimental traces and Figs. 2 and 4 report the plots whose extracted parameters are provided in the respective tables with their significance analysis.'

Thank you for clarifying. It would be helpful if the authors could include a statement in the legend to Figs. 2 and 4 and/or the tables to clarify that statistical outcomes pertaining to Figs. 2 and 4 are provided in the tables.

Requested statements are now included in the legends of Figs 2 and 4.

Dear Dr Pertici,

Re: JP-RP-2025-289622R3 "**Mechanics of blunting of actin-myosin interaction dynamics by the actinopathy-causing mutation E334Q in cytoskeletal γ -actin**" by Irene Pertici, Valentina Buonfiglio, Johannes N. Greve, Elena Battirosi, Duccio Fanelli, Dietmar J. Manstein, and Pasquale Bianco

We are pleased to tell you that your paper has been accepted for publication in The Journal of Physiology.

Yours sincerely,

Paul Greenhaff
Senior Editor
The Journal of Physiology

IMPORTANT POINTS TO NOTE FOLLOWING ACCEPTANCE OF YOUR PAPER:

- You can help your research get the attention it deserves! Check out Wiley's free Promotion Guide for best-practice recommendations for promoting your work at: www.wileyauthors.com/eoo/guide. You can learn more about Wiley Editing Services which offers professional video, design, and writing services to create shareable video abstracts, infographics, conference posters, lay summaries, and research news stories for your research at: www.wileyauthors.com/eoo/promotion.

- If you would like to receive our 'Research Roundup', a monthly newsletter highlighting the cutting-edge research published in The Physiological Society's family of journals (The Journal of Physiology, Experimental Physiology, Physiological Reports, The Journal of Nutritional Physiology and The Journal of Precision Medicine: Health and Disease), please click this link, fill in your name and email address and select 'Research Roundup': <https://www.physoc.org/journals-and-media/membernews>

EDITOR COMMENTS

Senior Editor:

Thank you for making the small changes requested to the figure legends. The manuscript is now acceptable for publication. Congratulations.

REFeree COMMENTS